# TINA: TINY REASONING MODELS VIA LoRA

**Shangshang Wang, Julian Asilis, Ömer Faruk Akgül, Enes Burak Bilgin,
Ollie Liu, Willie Neiswanger**

University of Southern California

`{shangsha,asilis,akgul,ebilgin,zliu2898,neiswang}@usc.edu`

## ABSTRACT

How cost-effectively can strong reasoning abilities be achieved in language models? Driven by this question, we present Tina, a family of tiny reasoning models achieved with high cost-efficiency. Tina shows that substantial reasoning performance can be developed using only minimal resources, by applying low-rank adaptation (LoRA) during reinforcement learning (RL), to an already tiny 1.5B parameter base model. This minimalist approach produces models that are competitive with, and sometimes surpass, SOTA RL reasoning models built upon the same base model. Crucially, this is achieved at a tiny fraction of the computational cost employed by existing models. In fact, the best Tina model achieves a >20% reasoning performance increase and 43.33% zero-shot Pass@1 accuracy on AIME24, at only $9 USD cost (i.e., an estimated 260x reduction). Our work reveals the surprising effectiveness of efficient RL reasoning via LoRA. We validate this across multiple open-source reasoning datasets and various ablation settings starting with a single, fixed set of hyperparameters. Furthermore, we explore the hypothesis that this effectiveness and efficiency stem from LoRA rapidly adapting the model to the structural format of reasoning rewarded by RL, while largely preserving the base model's underlying knowledge. In service of accessibility and open research, we fully open-source all code, training logs, model weights, and checkpoints.[1]

## 1 INTRODUCTION

Language models (LMs) demonstrate increasing proficiency across a variety of tasks, but achieving robust, multi-step reasoning remains a frontier challenge (Xu et al., 2025; Liu et al., 2025a). Enhancing complex reasoning via supervised fine-tuning (SFT) is a well-adopted technique, often utilizing a distillation process (Min et al., 2024; Huang et al., 2024) by which the model learns to mimic reasoning traces generated by more advanced models such as o1 (OpenAI, 2024) and R1 (DeepSeek-AI, 2025). This approach, while effective, can run the risk of instilling a shallow form of imitation in the learning model. In contrast, reinforcement learning (RL) enables models to learn directly from verifiable reward signals derived from curated data (DeepSeek-AI, 2025; Lambert et al., 2025). In doing so, RL can lead the model to explore a greater variety of logical paths and discover more robust solutions. However, RL pipelines are often complex and notoriously resource-intensive, typically involving substantial compute. This raises a fundamental question anchoring our research:

> How cost-effectively can one perform RL to efficiently instill reasoning abilities in LMs?

This question is motivated by the need to establish efficiency limits for RL reasoning, i.e., understanding the minimal requirements needed to achieve meaningful reasoning improvements. We focus on low-resource settings with tiny models and minimal parameter updates to probe the fundamental limits of instilling reasoning via RL, isolating training dynamics from sheer model scale. Although quantized small models offer an alternative path to efficient deployment, our aim is to understand how far RL-based reasoning can be pushed under tight compute budgets. (RUCAIBox STILL Team, 2025; Luo et al., 2025; Dang and Ngo, 2025) Moreover, understanding these efficiency limits addresses fundamental questions about the nature of reasoning in LMs. Specifically, recent studies

---

[1] `https://github.com/shangshang-wang/Tina`

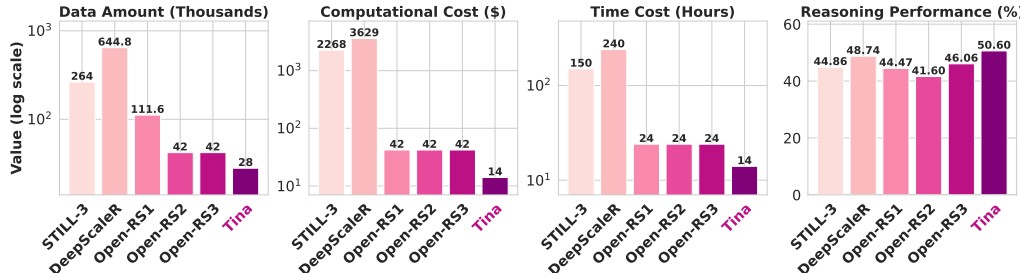

Figure 1: **Comparison between Tina and baseline models** Reasoning performance denotes the average score across AIME24/25, AMC23, MATH500, GPQA, and Minerva. The calculation of each comparative metric is detailed in Appendix A.1.

indicate that reasoning and knowledge storage are distinct capabilities: while knowledge capacity scales primarily with model size, reasoning performance may be less coupled to parameter count alone (Allen-Zhu and Li, 2024). This decoupling suggests that smaller models may possess untapped reasoning potential. Furthermore, evidence shows that parameter-efficient methods can adapt models for specific capabilities without degrading existing knowledge (Han et al., 2024), thereby offering a promising avenue for reasoning.

Therefore, in this paper we combine two key efficiency strategies: employing compact tiny base models (e.g., 1.5B) (DeepSeek-AI, 2025) and applying low-rank adaptation (LoRA) during RL. LoRA enables us to modify model behavior by training only a small fraction of parameters, making it an ideal technique for efficient reasoning. The synergy between these approaches forms the foundation of our Tina (Tiny Reasoning Models via LoRA) modeling framework. Importantly, this paper not only introduces the Tina models but also uses this minimalist setup to characterize reasoning improvements and explain why such a low-cost approach is effective. While our approach is a straightforward synthesis of LoRA and GRPO, the core contribution lies not in the method's complexity, but in the surprising discovery that substantial reasoning can be achieved at such a minimal cost, supported by an analysis of the training dynamics and a novel hypothesis to explain this effectiveness. We summarize our contributions as follows:

- **Surprising Effectiveness of Efficient RL Reasoning via LoRA** We show that our Tina models achieve performance competitive with, and in some cases even superior to, SOTA baseline models built on the same base model with full-parameter training, as shown in Figure 1 and in more detail in Table 2. In particular, the best Tina model achieves a >20% performance increase and 43.33% zero-shot Pass@1 accuracy on AIME24. Notably, the cost of reproducing the best Tina checkpoint stands at only **$9**, and of reproducing everything (all experiments, ablations, evaluations, etc.) presented in this paper *from scratch* at **$798**. We also additionally show FLOPS for Tina in Figure 2 and Appendix B.1.

- **Rapid Reasoning Format Adaptation Hypothesis** Based on our observations in RL post-training Tina, we hypothesize that LoRA's effectiveness and efficiency stem from rapidly adapting the reasoning format under RL while preserving base model knowledge—a likely more compute-efficient process than the deep knowledge integration of full-parameter training. Partial support comes from studies showing tiny LMs can reason effectively (Hugging Face, 2025; DeepSeek-AI, 2025), while large LMs can store broader world knowledge (Allen-Zhu and Li, 2025). This distinction suggests reasoning capabilities can be significantly enhanced by focusing on adapting the output format itself, consistent with our hypothesis about LoRA.

## 2 RELATED WORK

**Open-source reasoning replicas** Following the release of o1-preview (OpenAI, 2024), a number of open-source models have emerged to replicate or exceed its reasoning capabilities. STILL (Min et al., 2024) introduced a minimal yet high-quality training recipe designed to elicit reasoning with modest compute, demonstrating that imitation learning from curated traces remains competitive. Sky-T1 (NovaSky Team, 2025) further explored scaling using open instruction-tuned checkpoints, while SimpleRL (Zeng et al., 2025) highlighted the potential of lightweight RL without requiring large-scale

reward models. PRIME (Cui et al., 2025) and DeepScaleR (Luo et al., 2025) introduced process supervision and scaling experiments to isolate how reasoning quality evolves with model size and context length. s1 (Muennighoff et al., 2025) showed that even strong base models such as Qwen2.5-32B-Instruct benefit from fine-tuning on only 1k high-quality and long chain-of-thought data, which is curated to elicit reasoning capabilities. L1 (Aggarwal and Welleck, 2025) combined prompt engineering with data curation for RL, resulting in models that can efficiently and adaptively control their response length. Meanwhile, OREAL (Lyu et al., 2025) and OpenThinker (OpenThoughts Team, 2025) investigated self-correction and latent structure emergence through unsupervised and hybrid paradigms. The release of Open Reasoner Zero (Hu et al., 2025) and Open-RS (Dang and Ngo, 2025) further emphasized efficient RL-based strategies for reasoning with small models, completing a landscape of public alternatives for interpretability and reproducibility.

**RL with verifiable rewards** Reasoning tasks are well-suited to RL paradigms, as the correctness or quality of the final output often provides verifiable reward signals (*e.g.*, the validity of a logical deduction). Such signal can effectively guide the model towards learning more robust reasoning strategies. Consequently, various RL approaches have been explored within this domain. Certain methods introduce auxiliary reward models or critics to assess reasoning quality, such as ReFT (Luong et al., 2024) and REFINER (Paul et al., 2024). Other techniques employ explicit rule-based verification for self-correction (Wu et al., 2024). Some leverage self-play dynamics and exploration, such as mutual reasoning (Qi et al., 2024), or utilize inference-aware fine-tuning that optimizes performance under different sampling strategies (Chow et al., 2024). Notably, group relative policy optimization (GRPO) has been proposed as a variant of proximal policy optimization (PPO) which removes the need for a separate value network by using a group-based baseline for advantage estimation, improving training efficiency and leading to better reward alignment (Shao et al., 2024), as demonstrated by DeepSeek-R1 (DeepSeek-AI, 2025). Subsequently, Dr.GRPO (Liu et al., 2025b) introduced a subtle modification of GRPO addressing its bias to produce long responses.

## 3  METHOD

Tina is our family of models created by post-training the `DeepSeek-R1-Distill-Qwen-1.5B` base model using LoRA during RL (employing a GRPO-style algorithm). The "Tiny" designation encapsulates a deliberate focus on minimalism and efficiency across the entire framework. This encompasses not only the tiny base model architecture and the tiny parameter updates enabled by LoRA, but also extends to a tiny overall resource footprint. This minimized footprint is achieved through an efficient training pipeline leveraging accessible open-source resources, and requires only minimal hardware and budget resources (detailed in Section 4).

**GRPO formulation** Recall the following formulation of GRPO: For each question $q$, GRPO samples a group $G = \{o_1, o_2, \ldots, o_G\}$ of outputs from the old policy $\pi_{\theta_{\text{old}}}$ and optimizes the policy $\pi_\theta$ by maximizing the following objective:

$$\mathbb{E}_{\substack{q \sim P(Q), \\ \{o_i\}_{i=1}^G \sim \pi_{\theta_{\text{old}}}(O|q)}} \left[ \frac{1}{G} \sum_{i=1}^G \left( \min\left(\delta_i A_i, \text{clip}\left(\delta_i, 1-\epsilon, 1+\epsilon\right) A_i\right) - \beta \, \mathbb{D}_{\text{KL}}(\pi_\theta \| \pi_{\text{ref}}) \right) \right], \quad (1)$$

where we define $\delta_i = \frac{\pi_\theta(o_i|q)}{\pi_{\theta_{\text{old}}}(o_i|q)}$ and $A_i$ denotes the advantage computed from a group of rewards $\{r_1, r_2, \ldots, r_G\}$,

$$A_i = \frac{r_i - \text{mean}(\{r_1, r_2, \ldots, r_G\})}{\text{std}(\{r_1, r_2, \ldots, r_G\})},$$

and

$$\mathbb{D}_{\text{KL}}(\pi_\theta \| \pi_{\text{ref}}) = \frac{\pi_{\text{ref}}(o_i|q)}{\pi_\theta(o_i|q)} - \log \frac{\pi_{\text{ref}}(o_i|q)}{\pi_\theta(o_i|q)} - 1.$$

Note that $\epsilon$ and $\beta$ are parameters controlling the clipping range and KL penalty, respectively.

**LoRA formulation** While most existing open-source models that enable reasoning rely on the more expensive full-parameter training (Min et al., 2024; NovaSky Team, 2025; Zeng et al., 2025; Muennighoff et al., 2025; Aggarwal and Welleck, 2025; Cui et al., 2025; Luo et al., 2025; Lyu et al., 2025; OpenThoughts Team, 2025; Hu et al., 2025; Dang and Ngo, 2025), we investigate the use of LoRA for parameter-efficient post-training of reasoning models (Hu et al., 2021). Our goal

is to assess whether updating only a small fraction of parameters can still yield strong reasoning capabilities (Han et al., 2024). We follow the standard LoRA setup (Hu et al., 2021) that given a frozen pretrained weight matrix $W_0 \in \mathbb{R}^{d \times k}$ and trainable low-rank matrices $A \in \mathbb{R}^{d \times r}$ and $B \in \mathbb{R}^{r \times k}$ with $r \ll \min(d, k)$, the original forward pass $h(x) = W_0 x$ is modified to be

$$\hat{h}(x) = W_0 x + ABx.$$

In addition to its computational efficiency, LoRA provides modularity: by training only a low-rank decomposition of the parameter updates, it becomes possible to toggle reasoning behavior without maintaining multiple full model copies.

## 4   MODEL TRAINING DETAILS

### 4.1   TRAINING SETUP

We present our main training setup as follows. Please refer to Appendix A for additional details.

**Training datasets & evaluation benchmarks**  To facilitate meaningful comparisons and enable precise ablations, we post-train our Tina models via RL using the setups inherited from publicly available reasoning models. Tina models use open-source training recipes and training datasets from models like STILL-3 (RUCAIBox STILL Team, 2025), DeepScaleR (Luo et al., 2025), and Open-RS (Dang and Ngo, 2025). These models are also used as baselines. We evaluate the reasoning capabilities of our Tina models and the baselines across a diverse suite of six challenging benchmarks, primarily focused on mathematical and scientific reasoning: AIME24/25 (Art of Problem Solving, 2024), AMC23 (Art of Problem Solving, 2023), MATH500 (Hendrycks et al., 2021; Lightman et al., 2023), GPQA Diamond (Rein et al., 2024), and Minerva (Lewkowycz et al., 2022).

Table 1: **Computational cost breakdown**  Costs for all tasks in this paper, measured in USD. Our calculation is based on the full set of experiments shown in Appendix B.

| EXPERIMENTAL TASK | TRAINING COST EST. | EVALUATION COST EST. | TOTAL COST EST. |
|---|---|---|---|
| **Baseline Models Re-Evaluation** | - | $10 | $10 |
| **Robust Evaluation** | - | $110 | $110 |
| **Main: Tina-STILL-3-1.5B-preview** | $59 | $7 | $66 |
| **Main: Tina-DeepScaleR-1.5B-Preview** | $84 | $10 | $94 |
| **Main: Tina-Open-RS1** | $40 | $11 | $51 |
| **Main: Tina-Open-RS2** | $15 | $17 | $32 |
| **Main: Tina-Open-RS3** | $15 | $17 | $32 |
| **Ablation: OpenThoughts Dataset** | $84 | $10 | $94 |
| **Ablation: OpenR1 Dataset** | $59 | $7 | $66 |
| **Ablation: II-Thought Dataset** | $84 | $10 | $94 |
| **Ablation: LIMR Dataset** | $4 | $4 | $8 |
| **Ablation: Dr.GRPO Algorithm** | $15 | $17 | $32 |
| **Ablation: Learning Rate** | $7 | $8 | $15 |
| **Ablation: LoRA Rank/Alpha** | $14 | $16 | $30 |
| **Ablation: Format Reward Only** | $15 | $17 | $32 |
| **Ablation: Long Completion Length** | $15 | $17 | $32 |
| **Total: All Tasks** | **$510** | **$288** | **$798** |
| **Total: Main Tasks** | **$213** | **$62** | **$275** |
| **Total: Best Ckpt. in Each Main Task** | **$80** | **$5** | **$85** |
| **Total: All Ckpt. in Best-Performance Task** | **$14** | **$17** | **$31** |
| **Total: Best Ckpt. in Best-Performance Task** | **$8** | **$1** | **$9** |

**Computational cost breakdown**  We use a minimal setup with just two GPUs (NVIDIA L40S GPUs), accessible via commercial cloud platforms at an approximate rate of $1 USD per GPU hour, including 300 GB storage, based on pricing observed at the time of writing (Cudo Compute). The RL training process for our LoRA models proves to be highly efficient, with a single RL step typically

completing within one minute on this hardware. Evaluating a model checkpoint across our entire suite of six reasoning benchmarks requires approximately 1 L40S GPU hour on average. To ensure cost control, we initially established a conservative maximum budget of $100 USD for each complete experimental run, encompassing all stages from training to evaluation and miscellaneous tasks. As detailed in Table 1, our actual expenditures were significantly below this ceiling. Notably, all of our experiments required only minimal efforts on hyperparameter search, as detailed in Appendix A.3.

## 4.2 TRAINING TRICKS

Our training methodology for Tina models is designed to maximize cost-effectiveness and rapidly instill reasoning abilities, aligning with our core hypothesis that LoRA can efficiently adapt a tiny model to the structural format of reasoning (detailed in Section 6). Towards this goal, the following strategies were employed, utilizing a single, fixed set of hyperparameters across experiments to underscore robustness and minimize tuning overhead:

- **Concise reasoning with restricted completion length** To encourage the model to quickly learn the essence of effective reasoning—namely, achieving accurate results through concise and well-structured reasoning paths—we restrict the maximum completion length to 3,000 tokens during training for all Tina models. This not only guides the model towards more efficient problem-solving expressions but also reduces the computational load per training instance, further enhancing overall training speed and cost-efficiency. An ablation study on longer completion lengths is discussed in Section 6.2.

- **Accelerated adaptation with rank-alpha ratio and learning rate scheduling** To facilitate rapid assimilation of the new reasoning format, we deviate from common LoRA configurations and learning rate schedules. Specifically:
  - We set the LoRA alpha to be four times the LoRA rank (*e.g.*, rank 32, alpha 128), rather than the conventional two times. This amplified alpha encourages the model to more strongly incorporate the adaptations learned by the LoRA parameters, effectively making it "lean towards" the new reasoning structures reinforced during RL.
  - We employ an unconventional learning rate schedule. Instead of scheduling the learning rate decay over the standard training horizon, we schedule it over twice the training horizon. This results in a comparatively larger learning rate being applied at each step within the actual training period, promoting faster adaptation of the LoRA parameters to the reward signals.

  These adaptation strategies are chosen to align with our goal of achieving significant reasoning improvements with minimal computational costs, and thus promote efficient learning.

## 5 EMPIRICAL RESULTS

### 5.1 SURPRISING EFFECTIVENESS OF EFFICIENT RL REASONING VIA LORA

The results presented in Table 2 demonstrate that significant reasoning performance can be achieved efficiently, yielding models that are competitive with, or outperform, relevant baselines despite the inherent resource constraints of using parameter-efficient tuning.[2]

In Table 2, all Tina models exhibit substantial reasoning aptitude, achieving average scores in the range of 48.16% to 50.60%. By default, our reported scores are zero-shot Pass@1 Mean@1 performance. We also conduct robust evaluation experiment in Appendix B.2 with zero-shot Pass@1 Mean@10 performance to show the robustness of our approach in this paper. Significantly, nearly all Tina models notably outperform their corresponding baseline average scores, indicating marked improvements instilled by the parameter-efficient RL. The `Tina-Open-RS2` model yielded the highest average performance observed at 50.60%. Furthermore, these strong results were achieved with very limited training durations, ranging from just 19% to 57% of a full training epoch, highlighting the efficiency and rapid adaptation enabled by the Tina approach. These findings strongly support our central

---

[2]Tables 2 and 3 adopt a consistent naming pattern where "Tina-`X`" denotes that our model is the LoRA counterpart of a baseline model `X` or is trained on a dataset `X` (possibly followed with an extra ablation setup). This can reflect the model origin and serve as a direct reference to the public checkpoints for reproducibility.

Table 2: **Tina model evaluation** Performance comparison between Tina models and corresponding full-parameter-trained SOTA models on six reasoning tasks. The value in the *Steps* column indicates the training steps of the best model checkpoint within one epoch, while the full model checkpoint evaluation is shown in Appendix B.3. The *Baseline* column represents the average score achieved by a baseline model with full-parameter RL (all details of baseline evaluations described in Appendix B.1).

| TINA MODEL | STEPS (% OF 1 EPOCH) | AIME24 | AIME25 | AMC23 | MATH500 | GPQA | MINERVA | AVG. | BASELINE |
|---|---|---|---|---|---|---|---|---|---|
| Tina-STILL-3-1.5B-preview | 53% | 36.67 | **30.00** | 77.50 | 84.60 | 33.33 | 26.84 | 48.16 | 44.86 |
| Tina-DeepScaleR-1.5B-Preview | 19% | **43.33** | 26.67 | 67.50 | 86.20 | **37.88** | 28.68 | 48.38 | **48.74** |
| Tina-Open-RS1 | 34% | **43.33** | 20.00 | 80.00 | 84.00 | 35.35 | 28.68 | 48.56 | 44.47 |
| Tina-Open-RS2 | 51% | **43.33** | 26.67 | 77.50 | **87.00** | 36.36 | **32.72** | **50.60** | 41.60 |
| Tina-Open-RS3 | 57% | 36.67 | 23.33 | **82.50** | 85.20 | 37.37 | 31.62 | 49.45 | 46.06 |

hypothesis: robust reasoning capabilities can be effectively and economically cultivated in small language models through the targeted application of LoRA and RL.

Table 3: **Tina ablation variants evaluation** Performance evaluation of Tina's ablation variants on six reasoning tasks. The value in the *Steps* column indicates the training steps of the best model checkpoint within one epoch, and the full model checkpoint evaluation is shown in Appendix B.3.

| ABLATION ON DATASETS | STEPS (% OF 1 EPOCH) | AIME24 | AIME25 | AMC23 | MATH500 | GPQA | MINERVA | AVG. |
|---|---|---|---|---|---|---|---|---|
| Tina-OpenR1 (93.7k) | 13% | 36.67 | 26.67 | 75.00 | 86.80 | 39.90 | 30.51 | 49.26 |
| Tina-OpenThoughts (66.1k) | 30% | 36.67 | 26.67 | 72.50 | 84.80 | **41.41** | **33.09** | 49.19 |
| Tina-II-Thought (53.3k) | 30% | 40.00 | 20.00 | **80.00** | 86.00 | 33.84 | 26.84 | 47.78 |
| Tina-DeepScaleR (40.3k) | 19% | 43.33 | 26.67 | 67.50 | 86.20 | 37.88 | 28.68 | 48.38 |
| Tina-STILL-3 (33k) | 53% | 36.67 | **30.00** | 77.50 | 84.60 | 33.33 | 26.84 | 48.16 |
| Tina-Open-S1 (18.6k) | 34% | 43.33 | 20.00 | **80.00** | 84.00 | 35.35 | 28.68 | 48.56 |
| Tina-Open-RS (7k) | 51% | 43.33 | 26.67 | 77.50 | **87.00** | 36.36 | 32.72 | **50.60** |
| Tina-LIMR (1.39k) | 58% | **46.67** | 20.00 | 75.00 | 83.80 | 34.85 | 30.51 | 48.47 |
| **ABLATION ON LEARNING RATE** | **STEPS (% OF 1 EPOCH)** | **AIME24** | **AIME25** | **AMC23** | **MATH500** | **GPQA** | **MINERVA** | **AVG.** |
| Tina-LIMR-5e-6-lr | 29% | 36.67 | **26.67** | 75.00 | 83.60 | **35.86** | 29.41 | 47.87 |
| Tina-LIMR-1e-6-lr | 58% | **46.67** | 20.00 | 75.00 | 83.80 | 34.85 | **30.51** | **48.47** |
| Tina-LIMR-5e-7-lr | 58% | 43.33 | 16.67 | **77.50** | **84.60** | 34.85 | **30.51** | 47.91 |
| **ABLATION ON LoRA RANK** | **STEPS (% OF 1 EPOCH)** | **AIME24** | **AIME25** | **AMC23** | **MATH500** | **GPQA** | **MINERVA** | **AVG.** |
| Tina-LIMR-64-LoRA-rank | 29% | 20.00 | 30.00 | 77.50 | **84.20** | 38.38 | **31.62** | 46.95 |
| Tina-LIMR-32-LoRA-rank | 58% | **46.67** | 20.00 | 75.00 | 83.80 | 34.85 | 30.51 | 48.47 |
| Tina-LIMR-16-LoRA-rank | 58% | 43.33 | **33.33** | 70.00 | 83.20 | 35.35 | 28.31 | **48.92** |
| Tina-LIMR-8-LoRA-rank | 29% | 30.00 | 26.67 | 82.50 | 83.80 | 33.84 | 30.51 | 47.89 |
| Tina-LIMR-4-LoRA-rank | 86% | 36.67 | 20.00 | **85.00** | 83.80 | 31.82 | 29.04 | 47.72 |
| **ABLATION ON RL ALGORITHM** | **STEPS (% OF 1 EPOCH)** | **AIME24** | **AIME25** | **AMC23** | **MATH500** | **GPQA** | **MINERVA** | **AVG.** |
| Tina-Open-RS3-GRPO | 57% | 36.67 | **23.33** | **82.50** | **85.20** | 37.37 | **31.62** | 49.45 |
| Tina-Open-RS3-DrGRPO | **17%** | **43.33** | **23.33** | 80.00 | 85.00 | 35.35 | 30.15 | **49.53** |

## 5.2 ABLATION STUDY: KEY FACTORS

To better understand the factors influencing the performance and efficiency of our Tina models within the proposed low-cost framework, we conduct a series of ablation studies. These studies systematically investigate the impact of key design choices and hyperparameters: the underlying training dataset, the learning rate for LoRA updates, the rank of the LoRA adapters, and the specific RL algorithm employed. In each study, we typically vary one factor while holding others constant, often based on a high-performing configuration identified in our main experiments or preliminary runs. The results, summarized in Table 3, provide valuable insights of our economical approach.

- **Impact of training dataset** The first section of Table 3 highlights the influence of the dataset used for RL. We compared seven distinct datasets, varying significantly in size (from ≈1.4k to ≈94k samples). Strikingly, the `Tina-Open-RS` model, trained on a concise dataset of merely 7k examples, achieved the highest average score (50.60%). This outcome surpasses models trained on considerably larger datasets, such as `Tina-OpenR1` (93.7k samples, 49.26% avg).

This observation strongly supports our core "Tiny" premise and reflects the intuition that the quality and diversity of the dataset can matter more than the data size.

- **Impact of learning rate**  Using the `Tina-LIMR` configuration as a testbed (second section of Table 3), we assessed sensitivity to the learning rate. Among the tested values ($5 \times 10^{-6}$, $1 \times 10^{-6}$, and $5 \times 10^{-7}$), a learning rate of $1 \times 10^{-6}$ yielded the optimal average performance (48.47%) for this setup. While performance differences were not drastic, this indicates that learning rate selection remains a factor, although effective results were obtained without extensive tuning.

- **Impact of LoRA rank**  The third ablation study investigated the impact of LoRA rank, which directly controls the number of trainable parameters. Testing ranks 4, 8, 16, 32, and 64 on the `Tina-LIMR` setup, we observed considerable robustness. Ranks 8, 16, and 32 all produced strong results, with average scores clustering between 47.89% and 48.92%. Notably, rank 16 achieved the peak performance (48.92%) in this comparison, slightly outperforming rank 32 (48.47%). Performance decreased slightly at the extremes (rank 4 and 64). This study validates that highly parameter-efficient configurations (low ranks like 16 or 32) are effective, further enhancing the cost-effectiveness and minimal overhead of the Tina approach.

- **Impact of RL algorithm**  Finally, we compared two RL algorithms, GRPO and Dr.GRPO (Liu et al., 2025b), using the `Tina-Open-RS3` setup (final section of Table 3). Both algorithms led to similar peak average performance levels (49.45% for GRPO vs. 49.53% for Dr.GRPO). However, Dr.GRPO reached its best checkpoint significantly earlier in the training process (17% of an epoch vs. 57% for GRPO). This suggests potential advantages in sample efficiency for Dr.GRPO in this context with an alternative normalization in loss calculation, offering potentially faster convergence and further reductions in training time and cost.

## 6  EMERGING HYPOTHESIS

### 6.1  RAPID REASONING FORMAT ADAPTATION

To understand why LoRA facilitates both effective and efficient reasoning improvements via RL, we analyze the relationship between training compute and performance, alongside training dynamics.

**Less is more for reasoning**  As shown in Figure 2, plotting reasoning performance against approximate training FLOPs reveals a stark contrast between full-parameter and LoRA-based training regimes. We show FLOPS because they are more invariant than price for cost comparison. First, our LoRA-based Tina models achieve reasoning scores comparable or superior to fully fine-tuned baselines while requiring (in some cases) orders of magnitude fewer training FLOPs. We observe that in Tina models, increased training compute inversely affects performance, in contrast to full-parameter ones. This highlights a "less compute can yield more performance" phenomenon.

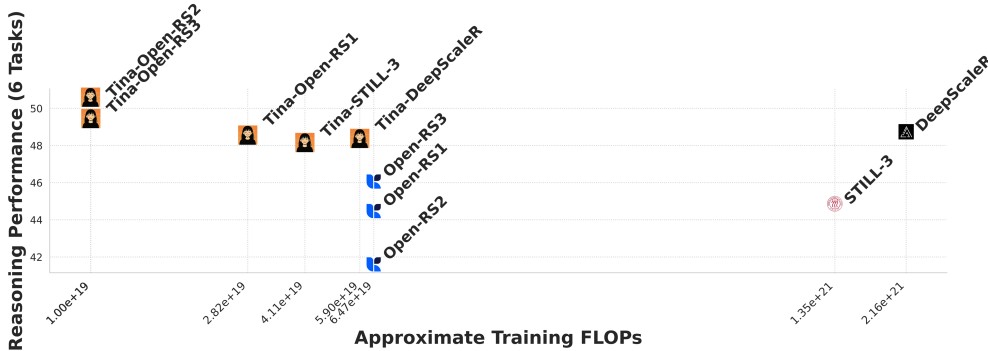

Figure 2: **Less is more for reasoning**  Approximate training FLOPs vs. reasoning performance comparison between Tina and baseline models. The calculation is detailed in Appendix A.1.

This finding supports our hypothesis regarding how LoRA is able to achieve such high efficiency, which relates to the principle of "learn structure/format, maintain knowledge." We posit that LoRA

excels in this scenario because RL for reasoning heavily rewards the model's ability to generate outputs in a specific, verifiable format or structure (*e.g.,* step-by-step reasoning chains). LoRA appears to be highly adept at learning these structural and stylistic patterns with minimal parameter changes, thus requiring very few FLOPs. At the same time, because LoRA modifies only a tiny fraction of the weights, it largely preserves the base model's vast pre-trained knowledge. Therefore, LoRA efficiently teaches the model how to format its existing knowledge into effective reasoning traces, rather than potentially imposing costly relearning of concepts or procedures that extensive full-parameter updates might entail. We hypothesize that this focus on structural adaptation allows Tina to achieve high reasoning performance with minimal computational investment.

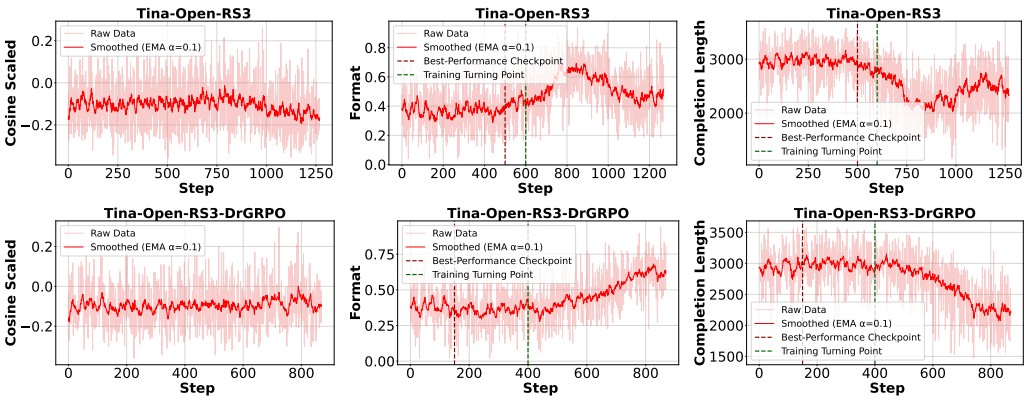

Figure 3: **Phase transition in LoRA-based RL** The "training turning point" in the legend means the step where the format-like metrics (*e.g.*, format reward, completion length) start to destabilize. Refer to Appendix C for the full set of plots for all Tina models. All raw data is from the Weights & Biases training logs and smoothed via exponential moving average (EMA) with factor 0.1.

**Phase transition** We use the term "phase transition" to describe a specific, observable training dynamic: a sharp, distinct turning point in format-related metrics (*e.g.*, format reward, completion length) that is decoupled from the more gradual evolution of the accuracy reward. As shown in Figure 3, we consistently observe such a training phase transition, or turning point, evident in the format-related metrics (format reward, *column 2*; completion length, *column 3*) across most Tina models, thus not a random success during training. Around this transition point (indicated by the green vertical dashed line), the format reward often peaks or destabilizes, while the completion length frequently reaches a maximum before potentially reversing its trend. Notably, this relatively sharp transition observed in format and length metrics does not typically have a corresponding distinct turning point in the accuracy reward plots (*column 1*). The accuracy reward often exhibits more gradual fluctuations or slower drift over the training duration, without a clear inflection aligned with the format transition.

Another crucial observation is the timing of optimal performance: the best-performing checkpoint, yielding the highest reasoning accuracy on held-out evaluations, consistently occurs just prior to or around this observed phase transition point in the format metrics. This decoupling between the dynamics of accuracy-based and format-based metrics suggests that the LoRA-based RL process rapidly optimizes the model's ability to adhere to the structural elements rewarded by the format score and length constraints. The subsequent transition point may signify where this structural optimization saturates, and becomes unstable. The fact that peak reasoning accuracy is achieved just before this format-driven transition implies that while learning the correct output format is essential and efficiently achieved via LoRA, pushing further on format-centric optimization alone does not necessarily yield better reasoning, and may even be detrimental.

## 6.2 ABLATION STUDY: COMPLETION LENGTH AND FORMAT REWARD

To further probe the "learn structure/format, maintain knowledge" effect and understand the dynamics of the observed "phase transition," we conducted targeted ablation studies.

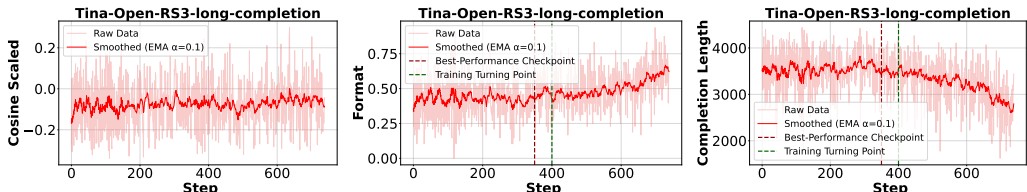

Figure 4: **Phase transition with long completion length** Training dynamics for `Tina-Open-RS3-long-completion` with a 10k maximum completion length.

**Long-completion training** One of the training tricks detailed in Section 4.2 is the restricted completion length (3k tokens), aiming to foster concise reasoning. To assess whether this restriction significantly influences the learning dynamics or inadvertently caps performance, we conducted an ablation where the maximum completion length was substantially increased to 10k tokens for a Tina model variant. As shown in Figure 4, increasing the completion length did not alter the training dynamics. The model's generated completion lengths during training peaked at approximately 4k tokens, remaining well below the 10k limit and comparable to the effective lengths observed with the default 3k setting. Crucially, the "phase transition" effect in format-related metrics persisted, mirroring the patterns observed in Figure 3. The accuracy reward also followed a similar trajectory. The persistence of the phase transition indicates that this phenomenon is a robust characteristic of the LoRA-driven format adaptation process, rather than an artifact of a restrictive length cap. Furthermore, the final reasoning evaluation scores of this model variant are comparable to those achieved with the 3k limit, *i.e.*, now 350 steps with 50.63 average score v.s. previously 500 steps with 49.45 average score (detailed in Appendix B.3).

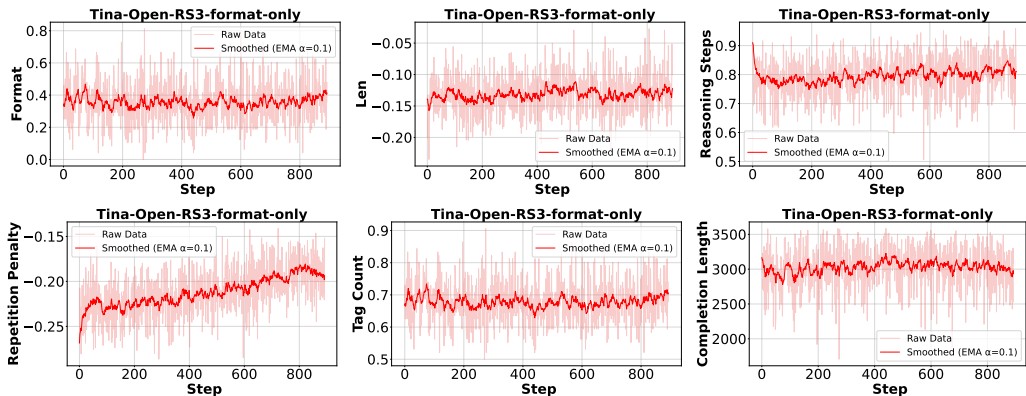

Figure 5: **Absence of phase transition in format-only training** Training dynamics for `Tina-Open-RS3-format-only` trained solely on format-related rewards.

**Format-only training** Our hypothesis posits that LoRA excels at adapting the model to the format of reasoning that is rewarded by RL, particularly when this format is linked to reasoning accuracy. To dissect the roles of format and accuracy rewards, we conducted an ablation study where a Tina model variant was trained using only format-based reward components, with the accuracy-based reward entirely removed. Illustrated in Figure 5, the "phase transition" disappeared in this format-only training regime. While the individual format-related reward components and completion length still evolved, the sharp turning point or destabilization signifying the phase transition was absent. The disappearance of the phase transition suggests that this dynamic is an emergent property of the interplay between LoRA's rapid adaptation to format requirements and the simultaneous drive to achieve reasoning accuracy. When the accuracy incentive is removed, LoRA may still adapt to the specified format objectives, but this adaptation becomes ineffective from the goal of producing correct reasoning. Specifically, it takes longer training time to reach similar reasoning performance, *i.e.*, now 850 steps with 50.56 average score v.s. previously 500 steps with 49.45 average score. This underscores that while LoRA is highly efficient at learning structural patterns, this capability is harnessed when guided by the objective of accurate problem-solving (detailed in Appendix B.3).

# 7 CONCLUSION

We presented Tina models to show that effective reasoning capabilities can be instilled in language models with high cost-efficiency. By combining LoRA with RL, we showed it is possible to achieve reasoning performance competitive with larger models that necessitate far more costly full-parameter training. We posited that LoRA's effectiveness stems from its ability to rapidly adapt the structural format of reasoning rewarded by RL, while largely preserving the knowledge embedded within the base model. This adaptation of reasoning pathways appears to be a more compute-efficient process.

## ACKNOWLEDGMENTS

The work was supported in part by the National Science Foundation (NSF) under Award No. 2427856. We also want to express our gratitude to the broader open-source community. This research was made possible by leveraging numerous publicly available resources, including training and evaluation framework, open datasets, accessible pre-trained language models, and the insights shared through technical reports. The computational resources required for the experiments described herein were provided by the Center for Advanced Research Computing (CARC) at USC.

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

# APPENDIX

## A  ADDITIONAL EXPERIMENTAL DETAILS

### A.1  TRAINING BUDGET: COST BREAKDOWN

We provide further details on how training data amounts, computational cost, time cost, and performance metrics reported in this paper – particularly those presented in figures like Figures 1 and 2 – were determined and should be interpreted.

**Overall comparison (Figure 1)**  For the baseline models included in Figure 1, the approximate training data amounts, computational costs (typically reported as GPU hours or total FLOPs), and training times are sourced from their respective technical reports or publications, leveraging the helpful summary provided in the Open-RS paper (Dang and Ngo, 2025). Reasoning performance scores for all models, encompassing both baselines and our Tina models, stem from results presented in Tables 2 and 6.

Also, it is crucial to understand the scope of reported costs:

- Epoch vs. Best Checkpoint: Costs cited for Tina and baseline models reflect the resources needed to complete a full training epoch or a predefined training run, not necessarily the minimal cost to reach the single best-performing checkpoint within that run.

- Training vs. Evaluation: Reported costs cover training only, omitting the computational expense required for model evaluation across benchmarks since such information is missing from several baseline models.

Particularly, the $9 USD in the abstract represents the estimated cost to train the Tina model up to its best-performing checkpoint and subsequently evaluate that specific checkpoint. For context comparing potential full training runs, the cost to train a Tina model for a complete epoch is $14 USD (training only). Including evaluation costs for such a full run would increase the total to approximately $31 USD. We emphasize the $9 as representing the efficient path to the best Tina model.

**FLOPs estimation (Figure 2)**  The approximate training FLOPs shown in Figure 2 serve as a hardware-agnostic measure of computational work. For both Tina and baseline models, these values were estimated based on reported training durations and hardware configurations sourced from technical reports or the Open-RS summary, using standard FLOPs calculation methodologies.

## A.2 TRAINING SETUP

**Baselines & Datasets.** Tina models inherit training recipes (e.g., hyperparameter and reward design) and training datasets all from public reasoning models. These models are used as baselines in the paper. All Tina and baseline models adopt `DeepSeek-R1-Distill-Qwen-1.5B` as their base model checkpoint with default open-source weights.

- **STILL-3-1.5B-preview** (RUCAIBox STILL Team, 2025) is a slow-thinking reasoning model developed through iterative RL on a curated dataset of 33k reasoning traces. The data originates from mathematics competitions and includes problems from MATH (Hendrycks et al., 2021; Lightman et al., 2023), NuminaMathCoT (LI et al., 2024), and AIME (1983–2023) (Art of Problem Solving, 2024). `Tina-STILL-3-1.5B-preview` uses the same dataset and reward pipeline.

- **DeepScaleR-1.5B-Preview** (Luo et al., 2025) focuses on long-context mathematical reasoning via RL, and is trained over approximately 40k problem-answer pairs drawn from the AIME (Art of Problem Solving, 2024), AMC (Art of Problem Solving, 2023), OMNI-MATH (Gao et al., 2024a), and STILL (RUCAIBox STILL Team, 2025) datasets. `Tina-DeepScaleR-1.5B-Preview` uses this dataset and mirrors the reward design.

- **Open-RS1/2/3** (Dang and Ngo, 2025) are three models from the `Open-RS` project exploring reasoning performance in 1.5B models trained via RL. All Open-RS models are trained on small, high-quality datasets further curated from the s1 (Muennighoff et al., 2025) (*i.e.*, Open-S1) and DeepScaleR (Luo et al., 2025) (*i.e.*, Open-DeepScaleR) datasets. The Tina models (`Tina-Open-RS1/2/3`) replicate these setups, using identical data splits and reward scaffolding.

**Hyperparameter and Reward Design.** We initiated parameter selection by replicating key parameters from `OpenR1` (Hugging Face, 2025) and `OpenRS` (Dang and Ngo, 2025). For all experiments presented in this paper, we deliberately adopted the default or recommended hyperparameter configurations provided in their works. These settings were kept largely fixed across different runs (Table 4). For ablation studies, only the specific factor under investigation (e.g., learning rate, LoRA rank/alpha, RL algorithm) was varied (Table 5). This approach intentionally circumvents costly hyperparameter search procedures for our specific setup, ensuring negligible tuning overhead and focusing on the efficacy of the core LoRA-based RL methodology.

For the main Tina results (Section 5.1), only reward designs were adjusted to ensure fair comparison of LoRA-trained models with their full-parameter counterparts. Particularly, all the reward designs including Accuracy, Format, Length, Cosine, Tag Count, Reasoning Steps, Repetition Penalty, are defined and implemented by the OpenR1 code repository.[3]

We show our default choice of hyperparameter in Table 4 and the varied hyperparameter and reward design in Table 5 for all the LoRA-based RL experiments.

---

[3]https://github.com/huggingface/open-r1

Table 4: Common hyperparameter of Tina models.

| | |
|---|---|
| `Tina-STILL-3-1.5B-preview` | LoRA |
| `Tina-DeepScaleR-1.5B-Preview` | LoRA |
| `Tina-Open-RS{X}-{Y}` | LoRA |
| `Tina-LIMR-{Z}` | LoRA |
| `Tina-OpenR1` | LoRA |
| `Tina-OpenThoughts` | LoRA |
| LoRA Modules | query, key, value, dense |
| LoRA Rank | 32 |
| LoRA $\alpha$ | 128 |
| LoRA Dropout | 0.05 |
| Algorithm | GRPO |
| Optimizer | AdamW |
| Optimizer Momentum | $\beta_1, \beta_2 = 0.9, 0.999$ |
| Learning Rate | 1e-6 |
| LR Scheduler | Cosine with Min LR |
| Warmup Ratio | 0.1 |
| Precision | BF16-mixed |
| Gradient Accumulation Step | 4 |
| Total Train Batch Size | 32 |
| Epochs | 1 |
| Hardware | $2 \times$ NVIDIA L40S |
| Max Prompt Length | 512 |
| Max Completion Length | 3584 |
| Number of Generation | 4 |
| Vllm GPU Memory Utilization | 0.4 |
| Vllm Max Model Length | 4608 |

Table 5: Varied hyperparameter and reward design of Tina models where "-" means unchanged from the common settings in Table 4.

| Model | LoRA Rank | LoRA Alpha | LoRA Dropout | Algorithm | Learning Rate | Reward Type | Reward Weights |
|---|---|---|---|---|---|---|---|
| `Tina-STILL-3-1.5B-preview` | - | - | - | - | - | Accuracy, Length | 2, 1 |
| `Tina-DeepScaleR-1.5B-Preview` | - | - | - | - | - | Accuracy, Format | 2, 1 |
| `Tina-Open-RS3` | - | - | - | - | - | Cosine, Format | 2, 1 |
| `Tina-Open-RS3-DrGRPO` | - | - | - | DrGRPO | - | Cosine, Format | 2, 1 |
| `Tina-Open-RS2` | - | - | - | - | - | Accuracy, Format | 2, 1 |
| `Tina-Open-RS1` | - | - | - | - | - | Accuracy, Format | 2, 1 |
| `Tina-LIMR` | - | - | - | - | - | Accuracy, Format | 2, 1 |
| `Tina-LIMR-5e-6-lr` | - | - | - | - | 5e-6 | Accuracy, Format | 2, 1 |
| `Tina-LIMR-5e-7-lr` | - | - | - | - | 5e-7 | Accuracy, Format | 2, 1 |
| `Tina-LIMR-64-LoRA-rank` | 64 | 256 | - | - | - | Accuracy, Format | 2, 1 |
| `Tina-LIMR-16-LoRA-rank` | 16 | 64 | - | - | - | Accuracy, Format | 2, 1 |
| `Tina-LIMR-8-LoRA-rank` | 8 | 32 | - | - | - | Accuracy, Format | 2, 1 |
| `Tina-LIMR-4-LoRA-rank` | 4 | 16 | - | - | - | Accuracy, Format | 2, 1 |
| `Tina-OpenR1` | - | - | - | - | - | Accuracy, Cosine, Format, Length, Tag Count, Reasoning Steps, Repetition Penalty | 1, 1, 1, 1, 1, 1, 1 |
| `Tina-OpenThoughts` | - | - | - | - | - | Accuracy, Cosine, Format, Length, Tag Count, Reasoning Steps, Repetition Penalty | 1, 1, 1, 1, 1, 1, 1 |

## A.3 TRAINING INFRASTRUCTURE & EVALUATION

- **Hardware** A key element of our low-cost approach was minimizing the hardware footprint. While distributed RL training algorithms like GRPO often benefit from using three or more GPUs (*e.g.,* dedicating one GPU to an inference engine such as `vLLM` for faster sample generation), we deliberately targeted a minimal setup using only two NVIDIA L40S GPUs.[4] To enable this, we co-located the RL training process and the `vLLM` on the same two GPUs by constraining `vLLM`'s GPU memory usage. The training itself utilized data parallelism across both GPUs. While running inference and training concurrently on two GPUs might result in a longer wall-clock training time compared to a setup with dedicated inference GPUs, it significantly reduces the hardware requirement.

- **Codebase** Our implementation builds upon `OpenR1`,[5] a fully open reproduction of DeepSeek-R1 (DeepSeek-AI, 2025) which combines the `Accelerate` (Gugger et al., 2022) and `Trl` (von Werra et al., 2020) libraries and the DeepSpeed ZeRO optimization (Rajbhandari et al., 2019). It aims to transparently replicate and extend RL methods used for improving reasoning in language models, particularly focusing on aligning model behavior with reasoning-oriented objectives via verifiable reward signals. Our methodology inherits its scaffolding, training utilities, and reward interfaces.

We evaluate the reasoning capabilities of our Tina models and the baselines across a diverse suite of six challenging benchmarks, primarily focused on mathematical and scientific reasoning:

- **AIME24/25** (Art of Problem Solving, 2024) contains 30 high-school-level math problems in algebra, geometry, number theory, and combinatorics from the 2024/2025 American Invitational Mathematics Examination. Each problem demands precise multi-step reasoning.

- **AMC23** (Art of Problem Solving, 2023) includes 40 problems from the 2023 American Mathematics Competition, offering a mix of logic and symbolic manipulation tasks.

- **MATH500** (Hendrycks et al., 2021; Lightman et al., 2023) is a benchmark comprising 500 competition mathematics problems derived from various sources, covering different difficulty levels and often necessitating multi-step derivation and calculation.

- **GPQA Diamond** (Rein et al., 2024), hereafter referred to as GPQA, consists of 198 PhD-level science questions across biology, chemistry, and physics. Each question is multiple-choice with subtle distractors.

- **Minerva** (Lewkowycz et al., 2022) includes 272 quantitative reasoning problems generally at the undergraduate level. The questions span multiple STEM fields, including physics, biology, chemistry, and economics, often requiring mathematical modeling or calculation steps. Includes tasks such as calculating enzyme kinetics from reaction data.

---

[4]Occasionally, NVIDIA RTX 6000 Ada GPUs were used instead, which is reflected in the system configuration metadata on Weights & Biases. From our practical experience, these two GPU types are similar in terms of cost and computational performance. For consistency, we report costs and compute metrics based on the L40S.

[5]`https://github.com/huggingface/open-r1`

# B  ADDITIONAL EMPIRICAL RESULTS

## B.1  BASELINE MODELS PERFORMANCE EVALUATION

For existing SOTA reasoning models, we note that performance scores reported in the literature for relevant models often stem from evaluations using disparate frameworks (*e.g.,* `verl` (Sheng et al., 2025), `lighteval` (Fourrier et al., 2023), `lm-eval-harness` (Gao et al., 2024b)) and inconsistent inference settings (such as different generation hyperparameters or varying numbers of GPUs). These variations can influence reported metrics, creating potential inconsistencies and hindering comparisons between models.

Table 6: Baseline models evaluation (all scores are zero-shot Pass@1 Mean@1).

| BASELINE MODEL | AIME24 | AIME25 | AMC23 | MATH500 | GPQA | MINERVA | AVG. |
|---|---|---|---|---|---|---|---|
| DeepSeek-R1-Distilled-Qwen-1.5B | 23.33 | 16.67 | 62.50 | 82.60 | 31.82 | 30.15 | 41.18 |
| II-Thought-1.5B-Preview | 30.00 | 23.33 | 72.50 | 86.80 | 31.90 | 30.88 | 45.90 |
| STILL-3-1.5B-preview | 26.67 | 26.67 | 67.50 | 86.40 | 34.34 | 27.57 | 44.86 |
| DeepScaleR-1.5B-Preview | 36.67 | 26.67 | 77.50 | **87.80** | 31.82 | 31.99 | **48.74** |
| Open-RS1 | 26.67 | 20.00 | 72.50 | 83.60 | 35.35 | 28.68 | 44.47 |
| Open-RS2 | 26.67 | 13.33 | 62.50 | 85.40 | 34.85 | 26.84 | 41.60 |
| Open-RS3 | **43.33** | 20.00 | 67.50 | 83.00 | 33.84 | 28.68 | 46.06 |
| FastCurl-1.5B-Preview | 26.67 | 20.00 | **82.50** | 83.40 | 35.88 | 30.25 | 46.45 |
| L1-Exact | 30.00 | **30.00** | 75.00 | 85.40 | 33.33 | **33.82** | 47.93 |
| L1-Max | 20.00 | 23.33 | 67.50 | 84.60 | **37.88** | 33.09 | 44.40 |

To mitigate these confounding factors, we performed a comprehensive re-evaluation of key baseline models using a single, consistent methodology throughout this paper. All baseline evaluations reported herein utilize the `lighteval` framework integrated with the `vLLM` (Kwon et al., 2023) inference engine for efficient generation. For comparability with prior work such as `OpenR1`, we maintained a fixed hardware configuration (two L40S GPUs) and applied a standardized set of `vLLM` inference parameters across all evaluated baseline models. All scores are zero-shot Pass@1 Mean@1 performance. The following is the evaluation command we use to combine `lighteval` and `vLLM` for performance evaluation on reasoning tasks. The `MODEL_PATH` should be replaced with either the local path or huggingface identifier to the model to be evaluated. `TASK` should be one of the six reasoning tasks including `aime24`, `aime25`, `amc23`, `math_500`, `gpqa:diamond`, and `minerva`. `PATH_TO_OPEN_R1_EVALUATE_SCRIPT` should be the path to the custom evaluate script provided by `OpenR1`.[6]

```
MODEL_ARGS="
pretrained=$MODEL_PATH,
dtype=bfloat16,
data_parallel_size=2,
max_model_length=32768,
gpu_memory_utilization=0.5,
generation_parameters={max_new_tokens:32768,temperature:0.6,top_p:0.95}"

lighteval vllm $MODEL_ARGS "custom|$TASK|0|0"
    --custom-tasks $PATH_TO_OPEN_R1_EVALUATE_SCRIPT
    --use-chat-template
```

---

[6]https://github.com/huggingface/open-r1/blob/4f5b21e21dec473af9729bce8e084deb16223ae4/src/open_r1/evaluate.py

## B.2 ROBUST EVALUATION EXPERIMENT

We also provide a robust evaluation experiment on the reasoning benchmarks. Specifically, we average the scores of AIME24/25 and AMC23 over 10 independent runs with different random seeds (i.e., they are zero-shot Pass@1 Mean@10 performance) and also add another benchmark, Olympiadbench (He et al., 2024). We run this robust experiment on all baseline models and our `Tina-DeepScaleR-1.5B-Preview` for demonstration. As shown in Tables 7 and 8, the performance perturbation on average is around 2% and does not affect our conclusion in the paper.

Table 7: Robust baseline models evaluation.

| BASELINE MODEL | AIME24 | AIME25 | AMC23 | MATH500 | GPQA | MINERVA | OLYMPIAD | AVG. | NON-ROBUST AVG. |
|---|---|---|---|---|---|---|---|---|---|
| DeepSeek-R1-Distilled-Qwen-1.5B | 29.00 | 21.67 | 71.50 | 82.60 | 31.82 | 30.15 | 53.63 | 45.77 | 41.18 |
| II-Thought-1.5B-Preview | 32.33 | 26.67 | 73.25 | 86.80 | 31.90 | 30.88 | 54.78 | 48.09 | 45.90 |
| STILL-3-1.5B-preview | 31.00 | **27.00** | 74.25 | 86.40 | 34.34 | 27.57 | 54.67 | 47.89 | 44.86 |
| DeepScaleR-1.5B-Preview | **37.00** | 25.67 | 68.25 | **87.80** | 31.82 | 31.99 | **56.00** | 48.36 | **48.74** |
| Open-RS1 | 28.67 | 26.00 | 75.00 | 83.60 | 35.35 | 28.68 | 53.33 | 47.23 | 44.47 |
| Open-RS2 | 31.33 | 20.33 | 70.00 | 85.40 | 34.85 | 26.84 | 53.93 | 46.10 | 41.60 |
| Open-RS3 | 30.00 | 25.67 | 68.75 | 83.00 | 33.84 | 28.68 | 49.48 | 45.63 | 46.06 |
| FastCurl-1.5B-Preview | 32.33 | 22.67 | **78.25** | 83.40 | 35.88 | 30.25 | **56.00** | **48.40** | 46.45 |
| L1-Exact | 23.67 | 21.00 | 73.50 | 85.40 | 33.33 | **33.82** | 52.89 | 46.23 | 47.93 |
| L1-Max | 24.33 | 21.00 | 73.00 | 84.60 | **37.88** | 33.09 | 53.04 | 46.71 | 44.40 |

Table 8: Robust evaluation of `Tina-DeepScaleR-1.5B-Preview`.

| CHECKPOINT STEPS (5039 STEPS PER EPOCH) | AIME24 | AIME25 | AMC23 | MATH500 | GPQA | MINERVA | OLYMPIAD | AVG. | NON-ROBUST AVG. |
|---|---|---|---|---|---|---|---|---|---|
| 500 | 30.00 | 24.00 | **74.50** | 82.40 | 39.39 | **31.25** | 53.93 | 47.93 | 45.65 |
| 1000 | **37.00** | 24.67 | 71.50 | **86.20** | 37.88 | 28.68 | 52.00 | **48.27** | **48.38** |
| 1500 | 30.33 | 23.67 | 71.75 | 84.80 | 32.83 | 29.41 | 50.81 | 46.23 | 46.17 |
| 2000 | 24.00 | 22.00 | 57.00 | 80.60 | 29.29 | 24.26 | 42.96 | 40.02 | 39.72 |
| 2500 | 18.67 | 16.00 | 55.00 | 75.00 | 31.31 | 18.01 | 41.33 | 36.47 | 34.47 |
| 3000 | 18.67 | 17.67 | 58.50 | 78.60 | 28.79 | 23.16 | 44.74 | 38.59 | 38.57 |
| 3500 | 23.67 | 22.00 | 60.75 | 80.40 | 31.82 | 24.26 | 44.59 | 41.07 | 40.94 |
| 4000 | 20.00 | 23.67 | 17.33 | 67.25 | **41.41** | 27.94 | 47.70 | 43.90 | 43.56 |
| 4500 | 23.33 | **27.00** | 19.67 | 66.00 | 34.85 | 26.47 | 49.04 | 43.40 | 42.99 |
| 5000 | 29.67 | 22.67 | 66.75 | 80.80 | 33.33 | 29.41 | 48.15 | 44.40 | 44.20 |

## B.3   ALL TINA MODELS PERFORMANCE EVALUATION

We present all Tina models' detailed evaluation performance during post-training across six reasoning tasks including AIME24/25, AMC23, MATH500, GPQA and Minerva.

Table 9: Evaluation of `Tina-STILL-3-1.5B-preview`.

| CHECKPOINT STEPS (3740 STEPS PER EPOCH) | AIME24 | AIME25 | AMC23 | MATH500 | GPQA | MINERVA | AVG. |
|---|---|---|---|---|---|---|---|
| 500 | 30.00 | 13.33 | 75.00 | 83.60 | 35.86 | **32.35** | 45.02 |
| 1000 | **36.67** | 20.00 | 65.00 | **84.80** | 32.32 | 27.94 | 44.46 |
| 1500 | 26.67 | 20.00 | 70.00 | 83.80 | **37.37** | 26.84 | 44.11 |
| 2000 | **36.67** | **30.00** | **77.50** | 84.60 | 33.33 | 26.84 | **48.16** |
| 2500 | 33.33 | **30.00** | 70.00 | 83.00 | 35.35 | 27.57 | 46.54 |
| 3000 | 30.00 | 20.00 | 67.50 | 82.60 | 30.81 | 25.74 | 42.78 |
| 3500 | 30.00 | 26.67 | 67.50 | 82.20 | 32.32 | 26.10 | 44.13 |

Table 10: Evaluation of `Tina-DeepScaleR-1.5B-Preview`.

| CHECKPOINT STEPS (5039 STEPS PER EPOCH) | AIME24 | AIME25 | AMC23 | MATH500 | GPQA | MINERVA | AVG. |
|---|---|---|---|---|---|---|---|
| 500 | 30.00 | 23.33 | 67.50 | 82.40 | 39.39 | **31.25** | 45.65 |
| 1000 | **43.33** | **26.67** | 67.50 | **86.20** | 37.88 | 28.68 | **48.38** |
| 1500 | 30.00 | 20.00 | **80.00** | 84.80 | 32.83 | 29.41 | 46.17 |
| 2000 | 20.00 | **26.67** | 57.50 | 80.60 | 29.29 | 24.26 | 39.72 |
| 2500 | 13.33 | 16.67 | 52.50 | 75.00 | 31.31 | 18.01 | 34.47 |
| 3000 | 26.67 | 16.67 | 57.50 | 78.60 | 28.79 | 23.16 | 38.57 |
| 3500 | 23.33 | 23.33 | 62.50 | 80.40 | 31.82 | 24.26 | 40.94 |
| 4000 | 20.00 | 20.00 | 70.00 | 82.00 | **41.41** | 27.94 | 43.56 |
| 4500 | 23.33 | 20.00 | 72.50 | 80.80 | 34.85 | 26.47 | 42.99 |
| 5000 | 20.00 | **26.67** | 75.00 | 80.80 | 33.33 | 29.41 | 44.20 |

Table 11: Evaluation of `Tina-II-Thought-1.5B-Preview`.

| CHECKPOINT STEPS (6660 STEPS PER EPOCH) | AIME24 | AIME25 | AMC23 | MATH500 | GPQA | MINERVA | AVG. |
|---|---|---|---|---|---|---|---|
| 500 | 33.33 | **23.33** | 77.50 | 83.20 | 31.31 | 27.57 | 46.04 |
| 1000 | 23.33 | 20.00 | 80.00 | 83.20 | 35.35 | 29.04 | 45.15 |
| 1500 | 40.00 | 20.00 | 80.00 | 86.00 | 33.84 | 26.84 | **47.78** |
| 2000 | 26.67 | 20.00 | **85.00** | 84.60 | 33.84 | 28.68 | 46.47 |
| 2500 | 30.00 | **23.33** | 75.00 | 85.00 | 40.40 | 26.47 | 46.70 |
| 3000 | 26.67 | 20.00 | 67.50 | **86.80** | 30.30 | 26.10 | 42.90 |
| 3500 | 33.33 | 16.67 | 67.50 | 84.00 | **40.91** | 30.88 | 45.55 |
| 4000 | 30.00 | 10.00 | 75.00 | 84.60 | 36.87 | 27.21 | 43.95 |
| 4500 | 26.67 | 16.67 | 72.50 | 85.40 | 33.84 | 25.37 | 43.41 |
| 5000 | 30.00 | **23.33** | 77.50 | 84.60 | 37.37 | **31.62** | 47.40 |

Table 12: Evaluation of `Tina-Open-RS3`.

| CHECKPOINT STEPS (875 STEPS PER EPOCH) | AIME24 | AIME25 | AMC23 | MATH500 | GPQA | MINERVA | AVG. |
|---|---|---|---|---|---|---|---|
| 50 | 26.67 | 23.33 | 75.00 | 84.20 | 37.37 | 29.04 | 45.94 |
| 100 | 30.00 | **30.00** | 65.00 | 83.00 | 37.37 | 29.78 | 45.86 |
| 150 | 36.67 | 16.67 | 65.00 | 84.80 | 27.78 | 27.94 | 43.14 |
| 200 | 20.00 | 26.67 | 70.00 | 83.80 | 33.33 | 27.94 | 43.62 |
| 250 | 36.67 | 20.00 | 65.00 | 84.60 | 38.38 | 28.31 | 45.49 |
| 300 | 33.33 | 26.67 | 70.00 | 85.20 | 30.81 | 30.15 | 46.03 |
| 350 | **40.00** | 16.67 | 77.50 | 84.40 | 39.90 | 27.94 | 47.74 |
| 400 | 30.00 | 16.67 | 70.00 | 82.80 | 35.86 | 31.25 | 44.43 |
| 450 | 36.67 | 26.67 | 70.00 | 85.60 | 33.84 | **32.72** | 47.58 |
| 500 | 36.67 | 23.33 | **82.50** | 85.20 | 37.37 | 31.62 | **49.45** |
| 550 | 26.67 | 16.67 | 80.00 | **86.00** | 35.35 | 29.78 | 45.75 |
| 600 | 30.00 | 26.67 | 70.00 | 84.60 | 37.88 | 29.78 | 46.49 |
| 650 | 20.00 | 23.33 | 80.00 | 85.00 | 33.33 | 27.94 | 44.93 |
| 700 | 33.33 | 13.33 | 72.50 | 85.00 | **40.40** | 31.99 | 46.09 |
| 750 | 33.33 | 23.33 | 75.00 | 83.60 | 31.31 | 27.57 | 45.69 |
| 800 | 30.00 | 23.33 | 65.00 | 84.20 | 38.38 | 29.04 | 44.99 |
| 850 | 26.67 | 26.67 | 75.00 | 83.80 | 31.82 | 27.94 | 45.32 |

Table 13: Evaluation of `Tina-Open-RS2`.

| CHECKPOINT STEPS (875 STEPS PER EPOCH) | AIME24 | AIME25 | AMC23 | MATH500 | GPQA | MINERVA | AVG. |
|---|---|---|---|---|---|---|---|
| 50 | 33.33 | 23.33 | **77.50** | 84.20 | 38.89 | 29.04 | 47.72 |
| 100 | 36.67 | 23.33 | 72.50 | 84.20 | 31.31 | 28.68 | 46.12 |
| 150 | 40.00 | 23.33 | 72.50 | 85.80 | 30.30 | 30.51 | 47.07 |
| 200 | 26.67 | 23.33 | 70.00 | 83.80 | 39.39 | 29.41 | 45.43 |
| 250 | **46.67** | 13.33 | 72.50 | 82.60 | 31.82 | 30.51 | 46.24 |
| 300 | 30.00 | **26.67** | 75.00 | 84.00 | 33.33 | 29.04 | 46.34 |
| 350 | 33.33 | 20.00 | 75.00 | 84.80 | 37.37 | 28.68 | 46.53 |
| 400 | 26.67 | 16.67 | 70.00 | 83.20 | 37.37 | 27.57 | 43.58 |
| 450 | 43.33 | **26.67** | **77.50** | **87.00** | 36.36 | **32.72** | **50.60** |
| 500 | 20.00 | 23.33 | 67.50 | 84.20 | 33.84 | 29.41 | 43.05 |
| 550 | 40.00 | 23.33 | 72.50 | 83.60 | **40.91** | 30.88 | 48.54 |
| 600 | 33.33 | 20.00 | 72.50 | 84.20 | 32.83 | 30.88 | 45.62 |
| 650 | 33.33 | 23.33 | 57.50 | 83.80 | 34.85 | 30.51 | 43.89 |
| 700 | 23.33 | **26.67** | 70.00 | 82.40 | 33.33 | 28.68 | 44.07 |
| 750 | 30.00 | 23.33 | 72.50 | 84.20 | 38.89 | 29.04 | 46.33 |
| 800 | 30.00 | **26.67** | 75.00 | 84.40 | 32.32 | 29.41 | 46.30 |
| 850 | 26.67 | 23.33 | 70.00 | 83.80 | 35.86 | 28.68 | 44.72 |

Table 14: Evaluation of `Tina-Open-RS1`.

| CHECKPOINT STEPS (2327 STEPS PER EPOCH) | AIME24 | AIME25 | AMC23 | MATH500 | GPQA | MINERVA | AVG. |
|---|---|---|---|---|---|---|---|
| 400 | 33.33 | 20.00 | 75.00 | 83.80 | 31.82 | 29.78 | 45.62 |
| 600 | 30.00 | **30.00** | 77.50 | 84.20 | 34.34 | **31.62** | 47.94 |
| 800 | **43.33** | 20.00 | 80.00 | 84.00 | 35.35 | 28.68 | **48.56** |
| 1000 | 33.33 | 20.00 | **82.50** | 84.40 | 35.86 | 29.78 | 47.64 |
| 1200 | 36.67 | 20.00 | 67.50 | **84.40** | **37.88** | 30.15 | 46.10 |
| 1400 | 30.00 | 20.00 | 67.50 | 83.40 | 31.82 | 29.78 | 43.75 |
| 1600 | 23.33 | 13.33 | 65.00 | 83.40 | 35.86 | 26.84 | 41.29 |
| 1800 | 26.67 | 20.00 | 75.00 | 84.20 | 34.34 | 27.57 | 44.63 |
| 2000 | 30.00 | 26.67 | 72.50 | 83.00 | 36.36 | 27.94 | 46.08 |
| 2200 | 30.00 | 23.33 | 70.00 | 81.40 | 30.81 | 26.47 | 43.67 |
| 2400 | 30.00 | 23.33 | 67.50 | 81.80 | 30.30 | 27.57 | 43.42 |

Table 15: Evaluation of `Tina-LIMR`.

| CHECKPOINT STEPS (174 STEPS PER EPOCH) | AIME24 | AIME25 | AMC23 | MATH500 | GPQA | MINERVA | AVG. |
|---|---|---|---|---|---|---|---|
| 50 | 20.00 | 26.67 | 67.50 | **85.40** | **37.88** | 30.51 | 44.66 |
| 100 | **46.67** | 20.00 | **75.00** | 83.80 | 34.85 | 30.51 | **48.47** |
| 150 | 26.67 | 20.00 | 72.50 | 84.00 | 37.37 | 30.15 | 45.12 |
| 200 | 33.33 | **30.00** | 62.50 | 83.40 | 29.80 | **30.88** | 44.99 |

Table 16: Evaluation of `Tina-OpenR1`.

| CHECKPOINT STEPS (11716 STEPS PER EPOCH) | AIME24 | AIME25 | AMC23 | MATH500 | GPQA | MINERVA | AVG. |
|---|---|---|---|---|---|---|---|
| 500 | 30.00 | 20.00 | **77.50** | 85.20 | 33.84 | 30.15 | 46.12 |
| 1000 | 30.00 | 23.33 | 72.50 | 85.60 | 33.84 | 26.67 | 45.32 |
| 1500 | **36.67** | 26.67 | 75.00 | **86.80** | **39.90** | 30.51 | **49.26** |
| 2000 | 26.67 | 23.33 | 67.50 | 83.20 | 29.80 | **31.62** | 43.69 |
| 2500 | 30.00 | 23.33 | 72.50 | 83.80 | 33.84 | 26.84 | 45.05 |
| 3000 | 20.00 | **30.00** | 67.50 | 84.60 | 34.34 | 28.31 | 44.13 |
| 3500 | **36.67** | 23.33 | 67.50 | 83.60 | 31.31 | 25.74 | 44.69 |

Table 17: Evaluation of `Tina-OpenThoughts`.

| CHECKPOINT STEPS (8259 STEPS PER EPOCH) | AIME24 | AIME25 | AMC23 | MATH500 | GPQA | MINERVA | AVG. |
|---|---|---|---|---|---|---|---|
| 500 | 33.30 | 16.67 | 77.50 | 84.20 | 35.86 | 30.15 | 46.28 |
| 1000 | 33.33 | 23.33 | **80.00** | 85.20 | 24.75 | 32.72 | 46.56 |
| 1500 | 30.00 | 23.33 | 70.00 | **86.00** | 37.88 | 29.04 | 46.04 |
| 2000 | 30.00 | 23.33 | 70.00 | 84.20 | 33.33 | 28.31 | 44.86 |
| 2500 | **36.67** | 26.67 | 72.50 | 84.80 | **41.41** | **33.09** | **49.19** |
| 3000 | 26.67 | 23.33 | 75.00 | 83.60 | 34.34 | 32.72 | 45.94 |
| 3500 | 20.00 | 16.67 | 60.00 | 84.20 | 32.32 | 26.10 | 39.88 |
| 4000 | 33.33 | 23.33 | 72.50 | 83.60 | 38.38 | 27.94 | 46.51 |
| 4500 | 30.00 | 20.00 | 65.00 | 85.00 | 33.84 | 26.84 | 43.45 |
| 5000 | 20.00 | **33.33** | 65.00 | 84.80 | 40.91 | 30.88 | 45.82 |

Table 18: Evaluation of `Tina-Open-RS3-DrGRPO`.

| CHECKPOINT STEPS (875 STEPS PER EPOCH) | AIME24 | AIME25 | AMC23 | MATH500 | GPQA | MINERVA | AVG. |
|---|---|---|---|---|---|---|---|
| 50 | 33.33 | 16.67 | 75.00 | 83.80 | 37.37 | 26.84 | 45.50 |
| 100 | 16.67 | 20.00 | 70.00 | 83.20 | 33.33 | 26.47 | 41.61 |
| 150 | **43.33** | 23.33 | **80.00** | 85.00 | 35.35 | 30.15 | **49.53** |
| 200 | 30.00 | 23.33 | 70.00 | 84.00 | **39.90** | 28.68 | 45.99 |
| 250 | 33.33 | **30.00** | 65.00 | 83.80 | 34.34 | 28.31 | 45.80 |
| 300 | 36.67 | 16.67 | 67.50 | 84.40 | 37.88 | 29.78 | 45.48 |
| 350 | 26.67 | **30.00** | 75.00 | 84.00 | 37.88 | 29.78 | 47.22 |
| 400 | 36.67 | 23.33 | 72.50 | 84.40 | 32.83 | 27.57 | 46.22 |
| 450 | 36.67 | 16.67 | 72.50 | **85.60** | 29.29 | 27.57 | 44.72 |
| 500 | 30.00 | 20.00 | 72.50 | **85.60** | 37.37 | 29.41 | 45.81 |
| 550 | 30.00 | 23.33 | 77.50 | 84.80 | 36.87 | **31.62** | 47.35 |
| 600 | 33.33 | 26.67 | 72.50 | 83.80 | 30.30 | 28.31 | 45.82 |
| 650 | 26.67 | 20.00 | 77.50 | 82.40 | 37.88 | 27.94 | 45.40 |
| 700 | 36.67 | 20.00 | **80.00** | 83.80 | 35.35 | 31.25 | 47.85 |
| 750 | 30.00 | 26.67 | 75.00 | 84.20 | 38.89 | 27.57 | 47.06 |
| 800 | 20.00 | **30.00** | 75.00 | 82.40 | 35.86 | 28.31 | 45.26 |
| 850 | 23.33 | 20.00 | 72.50 | 85.40 | 36.36 | 30.15 | 44.62 |

Table 19: Evaluation of `Tina-Open-RS3-long-completion`.

| CHECKPOINT STEPS (875 STEPS PER EPOCH) | AIME24 | AIME25 | AMC23 | MATH500 | GPQA | MINERVA | AVG. |
|---|---|---|---|---|---|---|---|
| 50 | 26.67 | 20.00 | 72.50 | **87.00** | 35.35 | 29.41 | 45.16 |
| 100 | 26.67 | 26.67 | 77.50 | 85.80 | 31.31 | 27.21 | 45.86 |
| 150 | 20.00 | 30.00 | 67.50 | 85.00 | 34.34 | 30.15 | 44.50 |
| 200 | 30.00 | 16.67 | 80.00 | 84.40 | 32.83 | 29.78 | 45.61 |
| 250 | 26.67 | 23.33 | 75.00 | 83.00 | 36.87 | 33.09 | 46.33 |
| 300 | 33.33 | 16.67 | 70.00 | 86.40 | 32.32 | 27.21 | 44.32 |
| 350 | **43.33** | 23.33 | **82.50** | 83.20 | **40.91** | 30.51 | **50.63** |
| 400 | 26.67 | 23.33 | 72.50 | 84.40 | 35.35 | 27.57 | 44.97 |
| 450 | 33.33 | 16.67 | 67.50 | 83.40 | 33.84 | **34.19** | 44.82 |
| 500 | 26.67 | 20.00 | 80.00 | 84.60 | 30.81 | 30.51 | 45.43 |
| 550 | 33.33 | 23.33 | 70.00 | 84.20 | 38.38 | 29.78 | 46.50 |
| 600 | 33.33 | 23.33 | 67.50 | 84.80 | 33.33 | 27.57 | 44.98 |
| 650 | 26.67 | 20.00 | 75.00 | 84.20 | 36.36 | 30.51 | 45.46 |
| 700 | 30.00 | **33.33** | 70.00 | 84.20 | 37.37 | 28.68 | 47.26 |
| 750 | 30.00 | 30.00 | 72.50 | 84.60 | 38.38 | 27.94 | 47.24 |
| 800 | 30.00 | 20.00 | 77.50 | 86.60 | 37.37 | 26.47 | 46.32 |
| 850 | 33.33 | 26.67 | 72.50 | 84.00 | 30.30 | 31.62 | 46.40 |

Table 20: Evaluation of `Tina-Open-RS3-format-only`.

| CHECKPOINT STEPS (875 STEPS PER EPOCH) | AIME24 | AIME25 | AMC23 | MATH500 | GPQA | MINERVA | AVG. |
|---|---|---|---|---|---|---|---|
| 50 | 40.00 | 20.00 | 77.50 | 85.80 | 32.32 | 29.04 | 47.44 |
| 100 | 26.67 | **30.00** | 70.00 | 84.20 | 31.82 | 27.21 | 44.98 |
| 150 | 30.00 | 23.33 | 72.50 | 84.60 | 35.35 | 29.41 | 45.87 |
| 200 | 30.00 | 16.67 | 75.00 | 84.60 | 31.82 | **33.46** | 45.26 |
| 250 | 26.67 | 20.00 | 70.00 | 84.20 | 32.83 | 28.31 | 43.67 |
| 300 | 33.33 | 23.33 | 70.00 | 84.80 | 27.27 | 29.04 | 44.63 |
| 350 | 36.67 | 16.67 | 67.50 | 85.60 | **38.38** | 27.21 | 45.34 |
| 400 | 40.00 | 20.00 | 57.50 | 85.80 | 33.33 | 27.21 | 43.97 |
| 450 | 30.00 | 26.67 | 77.50 | 84.20 | 29.29 | 29.41 | 46.18 |
| 500 | 30.00 | 16.67 | 67.50 | 83.60 | 32.32 | 30.88 | 43.50 |
| 550 | 33.00 | **30.00** | 80.00 | 84.80 | 32.32 | 27.21 | 47.89 |
| 600 | 26.67 | 20.00 | 75.00 | 86.00 | 34.85 | 28.31 | 45.14 |
| 650 | 40.00 | 23.33 | 75.00 | 85.40 | 36.36 | 27.94 | 48.01 |
| 700 | **43.33** | 26.67 | 70.00 | 84.00 | 34.34 | 29.78 | 48.02 |
| 750 | 33.33 | 23.33 | 77.50 | **86.20** | 30.81 | 30.51 | 46.95 |
| 800 | 30.00 | **30.00** | 75.00 | 84.60 | 36.36 | 33.09 | 48.18 |
| 850 | 40.00 | **30.00** | **82.50** | 84.60 | 35.35 | 30.88 | **50.56** |

Table 21: Evaluation of `Tina-LIMR-5e-6-lr` with learning rate `5e-6`.

| CHECKPOINT STEPS (174 STEPS PER EPOCH) | AIME24 | AIME25 | AMC23 | MATH500 | GPQA | MINERVA | AVG. |
|---|---|---|---|---|---|---|---|
| 50 | 20.00 | 26.67 | 67.50 | **85.40** | **37.88** | 30.51 | 44.66 |
| 100 | **46.67** | 20.00 | **75.00** | 83.80 | 34.85 | 30.51 | **48.47** |
| 150 | 26.67 | 20.00 | 72.50 | 84.00 | 37.37 | 30.15 | 45.12 |
| 200 | 33.33 | **30.00** | 62.50 | 83.40 | 29.80 | **30.88** | 44.99 |

Table 22: Evaluation of `Tina-LIMR-5e-7-lr` with learning rate `5e-7`.

| CHECKPOINT STEPS (174 STEPS PER EPOCH) | AIME24 | AIME25 | AMC23 | MATH500 | GPQA | MINERVA | AVG. |
|---|---|---|---|---|---|---|---|
| 50 | 40.00 | 13.33 | 72.50 | 83.00 | 34.34 | 29.04 | 45.37 |
| 100 | **43.33** | 16.67 | **77.50** | 84.60 | 34.85 | 30.51 | **47.91** |
| 150 | 30.00 | **23.33** | 72.50 | **86.20** | **37.37** | 30.51 | 46.65 |
| 200 | 33.33 | 13.33 | 70.00 | 83.20 | 29.29 | **31.25** | 43.40 |

Table 23: Evaluation of `Tina-LIMR-64-LoRA-rank` with LoRA rank 64 and alpha 512.

| CHECKPOINT STEPS (174 STEPS PER EPOCH) | AIME24 | AIME25 | AMC23 | MATH500 | GPQA | MINERVA | AVG. |
|---|---|---|---|---|---|---|---|
| 50 | 20.00 | **30.00** | **77.50** | 84.20 | **38.38** | **31.62** | **46.95** |
| 100 | 30.00 | 23.33 | 72.50 | **84.60** | 32.32 | 29.78 | 45.42 |
| 150 | **36.67** | 20.00 | 70.00 | 83.40 | 31.82 | 30.88 | 45.46 |
| 200 | 33.33 | 20.00 | 72.50 | 85.00 | 29.80 | 29.41 | 45.01 |

Table 24: Evaluation of `Tina-LIMR-16-LoRA-rank` with LoRA rank 16 and alpha 64.

| CHECKPOINT STEPS (174 STEPS PER EPOCH) | AIME24 | AIME25 | AMC23 | MATH500 | GPQA | MINERVA | AVG. |
|---|---|---|---|---|---|---|---|
| 50 | 33.33 | 23.33 | 62.50 | **84.20** | 38.89 | **31.25** | 45.58 |
| 100 | **43.33** | **33.33** | 70.00 | 83.20 | 35.35 | 28.31 | **48.92** |
| 150 | 26.67 | 16.67 | 72.50 | 83.40 | 35.35 | 29.04 | 43.94 |
| 200 | 36.67 | 20.00 | **75.00** | 83.00 | 39.39 | 30.51 | 47.43 |

Table 25: Evaluation of `Tina-LIMR-8-LoRA-rank` with LoRA rank 8 and alpha 32.

| CHECKPOINT STEPS (174 STEPS PER EPOCH) | AIME24 | AIME25 | AMC23 | MATH500 | GPQA | MINERVA | AVG. |
|---|---|---|---|---|---|---|---|
| 50 | 30.00 | **26.67** | **82.50** | 83.80 | 33.84 | 30.51 | **47.89** |
| 100 | 26.67 | 16.67 | 72.50 | 84.00 | 36.87 | 29.78 | 44.42 |
| 150 | **53.33** | 20.00 | 60.00 | 83.20 | **37.37** | **30.88** | 47.46 |
| 200 | 23.33 | 20.00 | 72.50 | **85.40** | 32.83 | 28.68 | 43.86 |

Table 26: Evaluation of `Tina-LIMR-4-LoRA-rank` with LoRA rank 4 and alpha 16.

| CHECKPOINT STEPS (174 STEPS PER EPOCH) | AIME24 | AIME25 | AMC23 | MATH500 | GPQA | MINERVA | AVG. |
|---|---|---|---|---|---|---|---|
| 50 | 30.00 | 23.33 | 65.00 | 85.00 | 35.35 | **29.78** | 44.74 |
| 100 | 26.67 | **26.67** | 72.50 | 82.80 | 34.85 | 29.04 | 45.42 |
| 150 | **36.67** | 20.00 | **85.00** | 83.80 | 31.82 | 29.0 | **47.72** |
| 200 | 33.33 | 23.33 | 77.50 | **85.40** | 35.86 | 28.31 | 47.29 |

## C  ALL TINA MODELS TRAINING PHASE TRANSITION

We present all Tina models' training phase transitions along the training dynamics. The raw data is from the Weights & Biases training logs and smoothed via exponential moving average (EMA) with factor $0.1$. Specifically, we observe clear transitions in `Tina-DeepScaleR-1.5B-Preview`, `Tina-STILL-3-1.5B-preview`, `Tina-II-Thought-1.5B-Preview`, `Tina-Open-RS1`, `Tina-Open-RS2`, `Tina-Open-RS3`, `Tina-Open-RS3-GRPO`, `Tina-Open-RS3-long-completion`, as shown in Figures 6, 7, 8, 9, and. For `Tina-OpenR1` and `Tina-Thoughts` (Figures 10 and 11), the observation is similar, except the best-performing checkpoint is achieved after the training turning point, rather than before.

However, we do not observe such a transition in all Tina variants on the LIMR dataset, as shown in Figures 12, 13, and 14, possibly because its small data size leads to training periods which are too brief to extract meaningful information. Also, we do not observe the transition in `Tina-Open-RS3-format-only` in Figure 15 due to the absence of accuracy rewards.

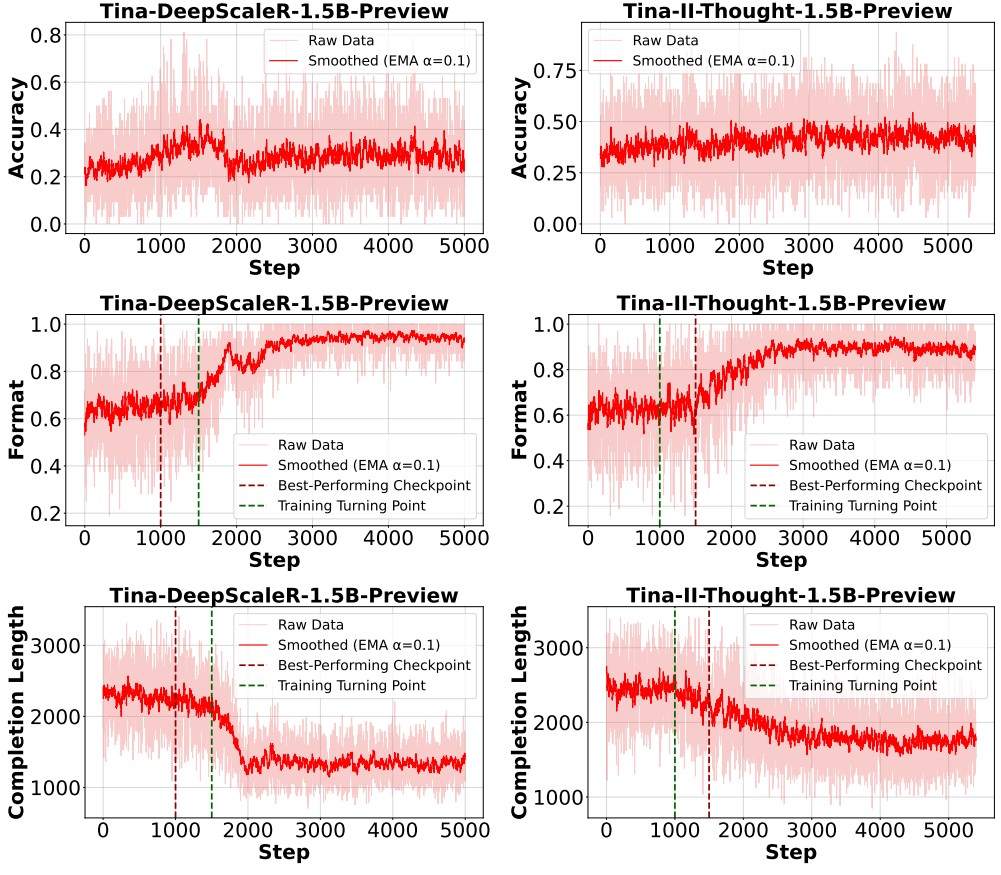

Figure 6: Transition in `Tina-DeepScaleR-1.5B-Preview` and `Tina-II-Thought-1.5B-Preview`.

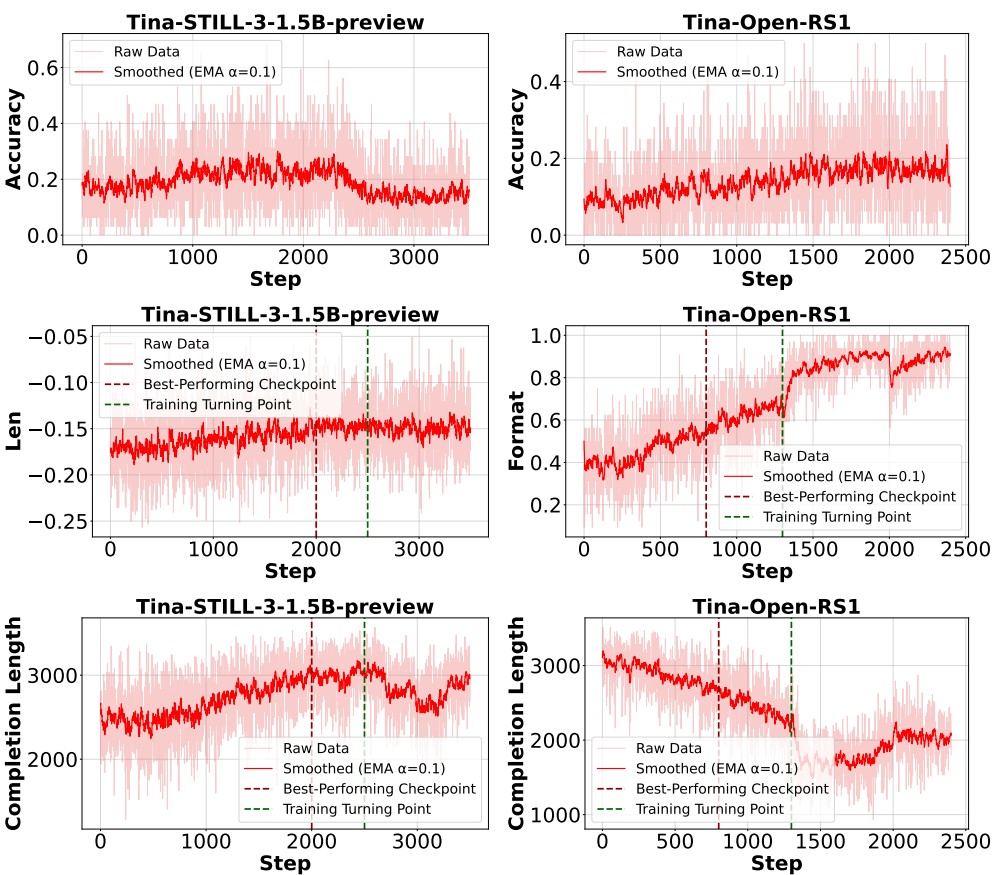

Figure 7: Transition in `Tina-STILL-3-1.5B-preview` and `Tina-Open-RS1`.

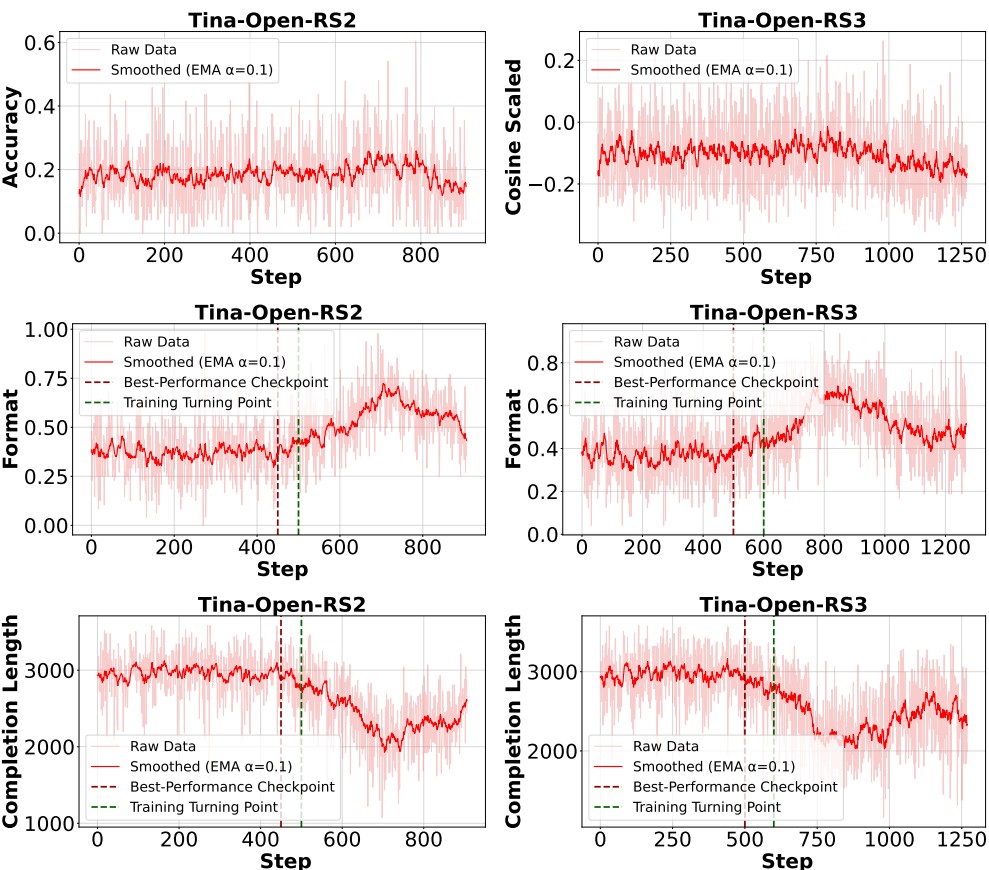

Figure 8: Transition in `Tina-Open-RS2` and `Tina-Open-RS3`.

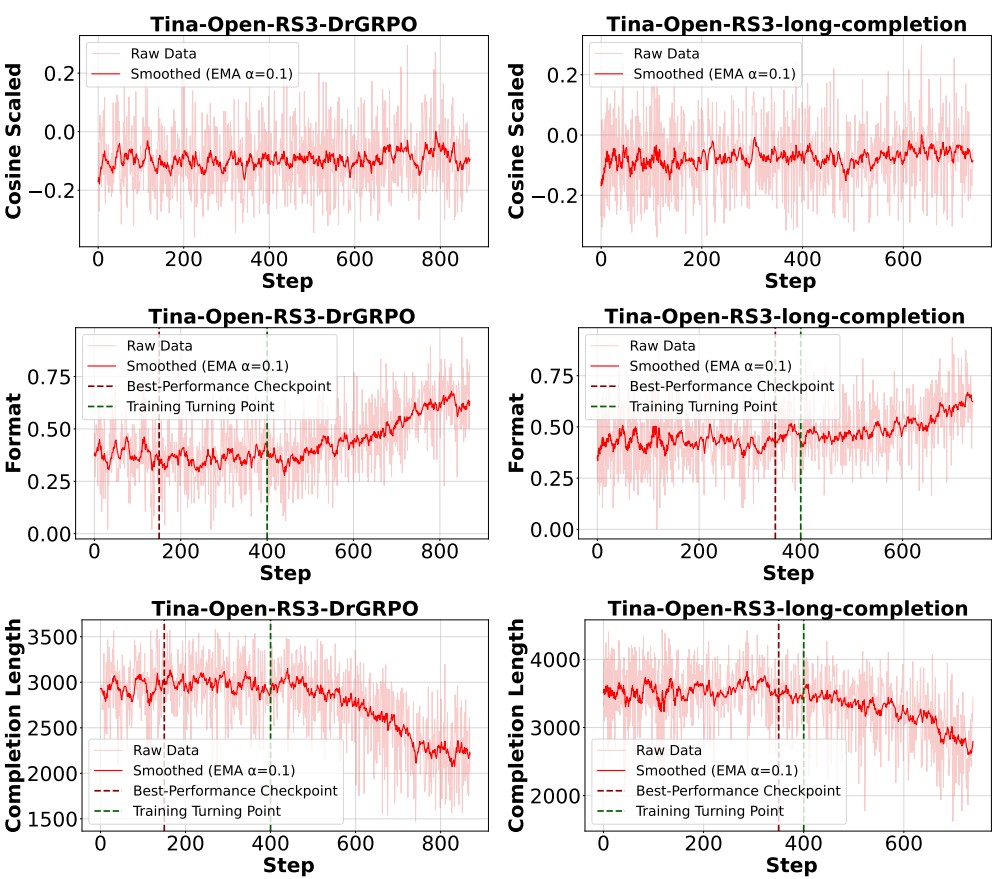

Figure 9: Transition in `Tina-Open-RS3-GRPO` and `Tina-Open-RS3-long-completion`.

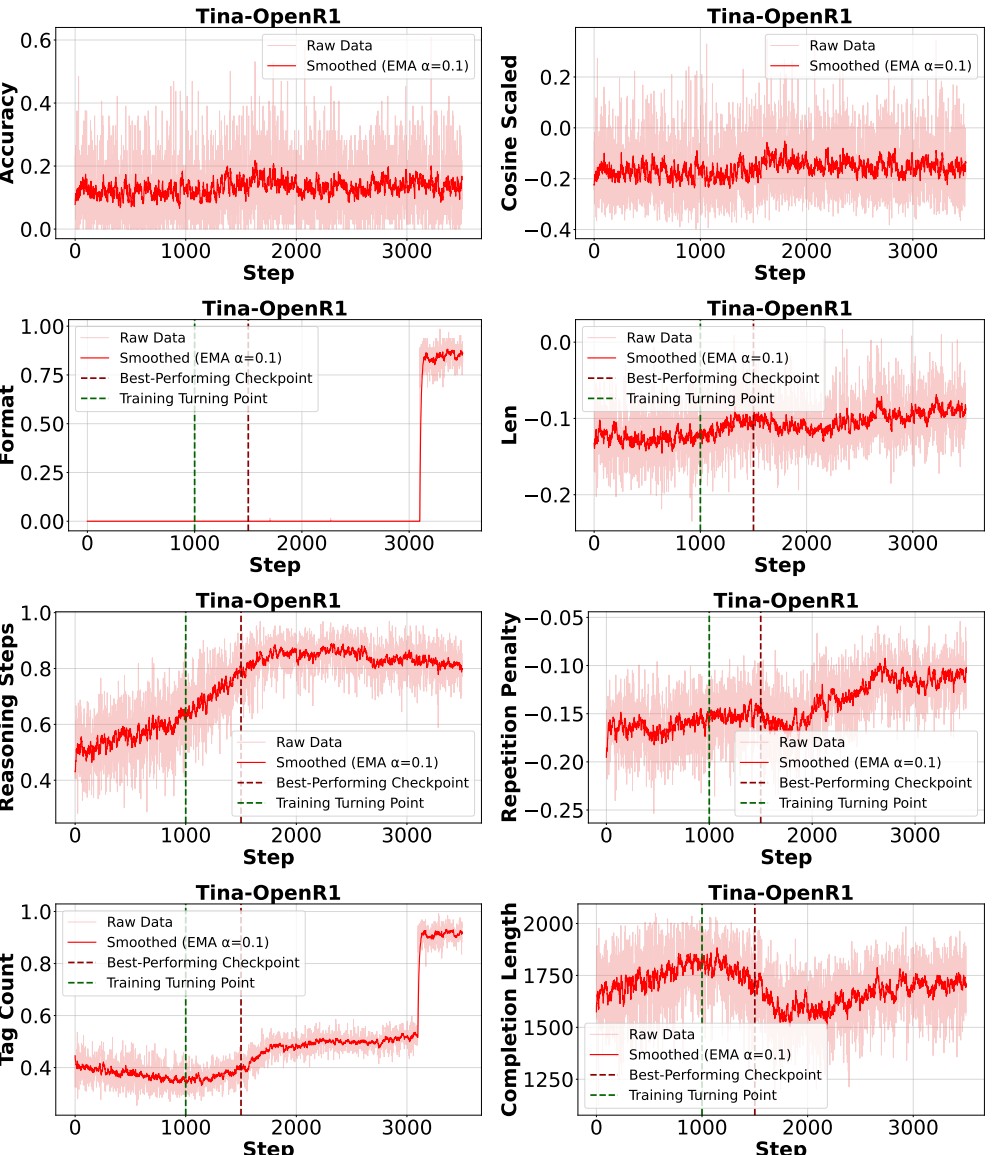

Figure 10: Transition in `Tina-OpenR1`.

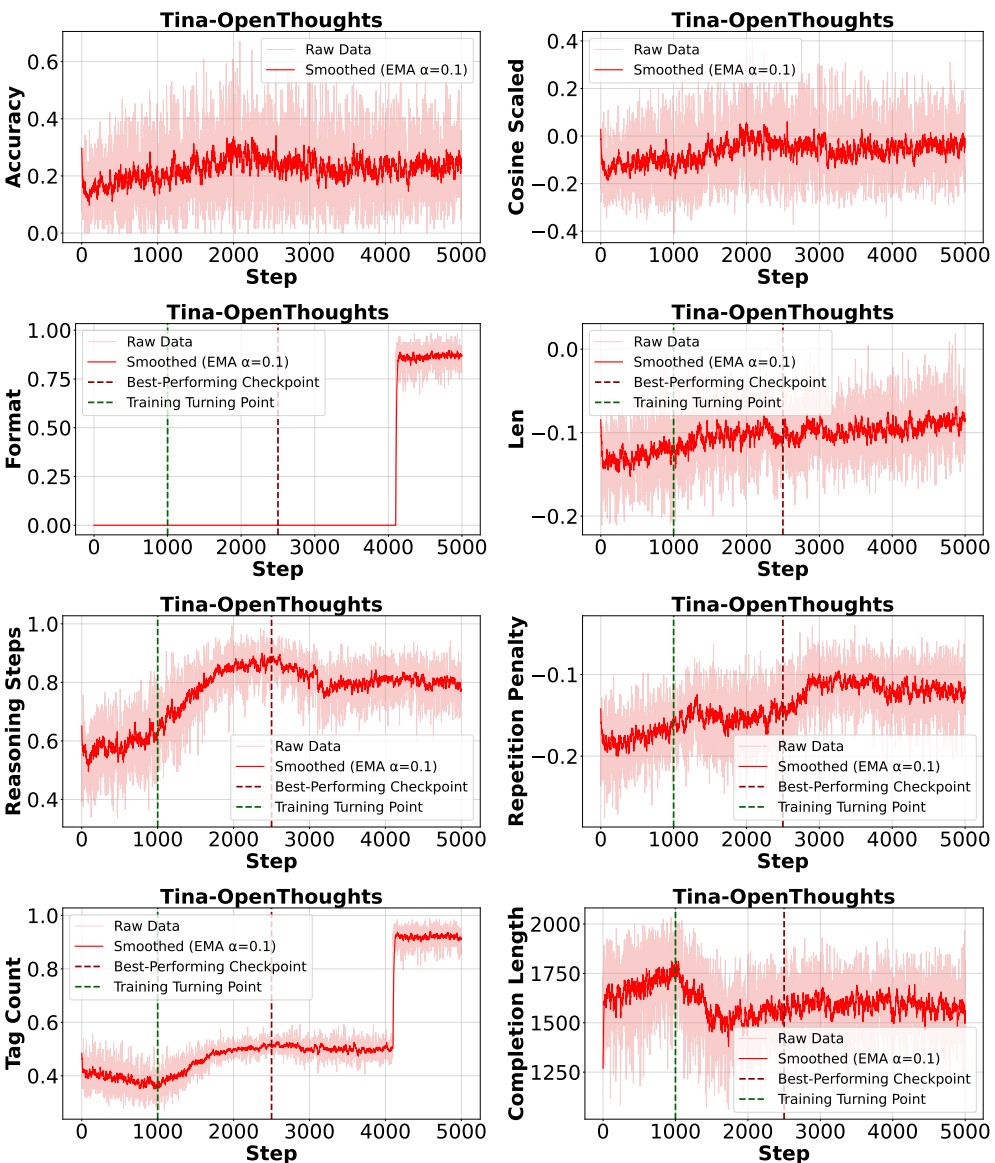

Figure 11: Transition in `Tina-OpenThoughts`.

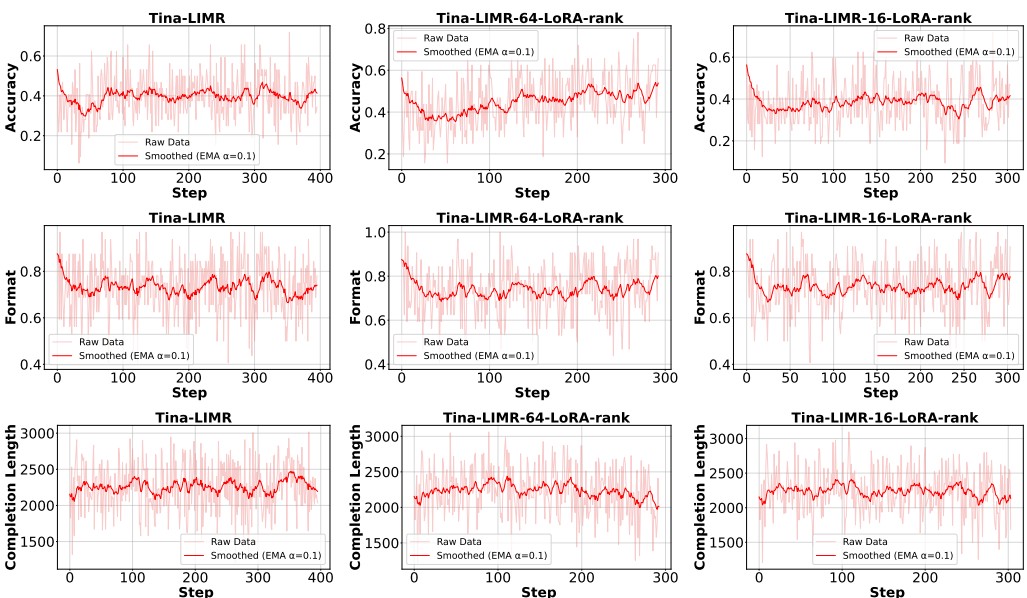

Figure 12: Transition in `Tina-LIMR`, `Tina-LIMR-64-LoRA-rank` and `Tina-LIMR-16-LoRA-rank`.

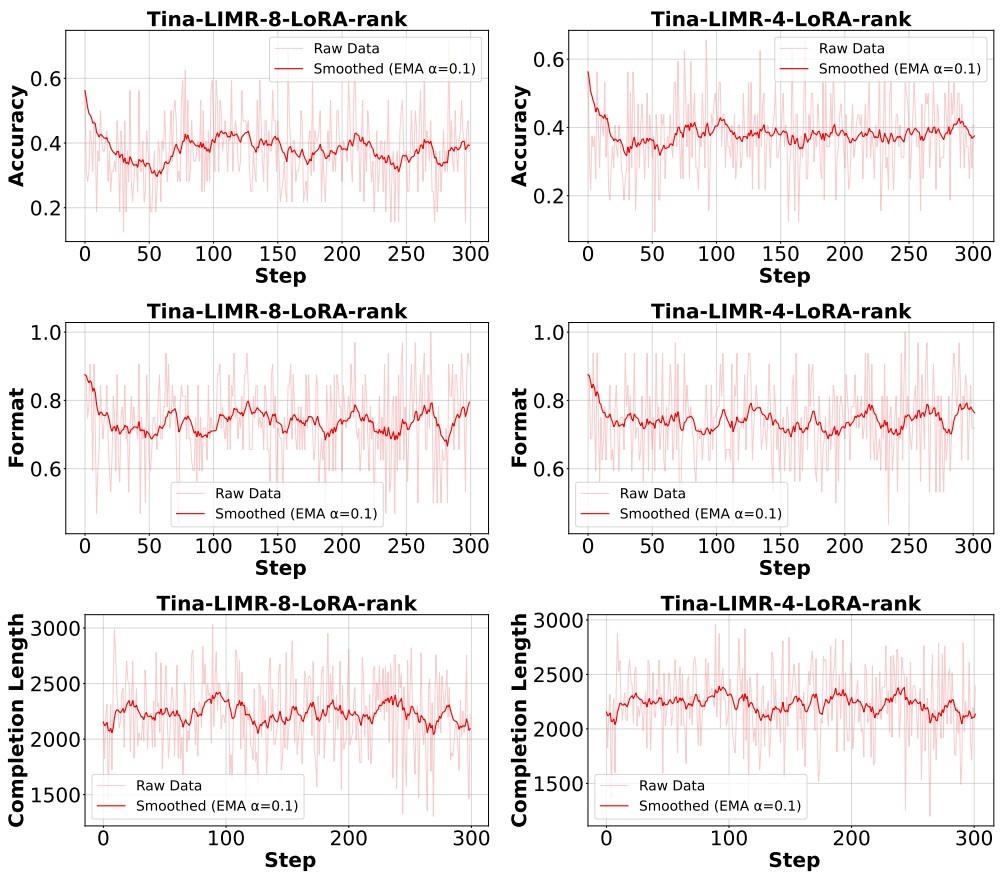

Figure 13: Transition in `Tina-LIMR-8-LoRA-rank` and `Tina-LIMR-4-LoRA-rank`.

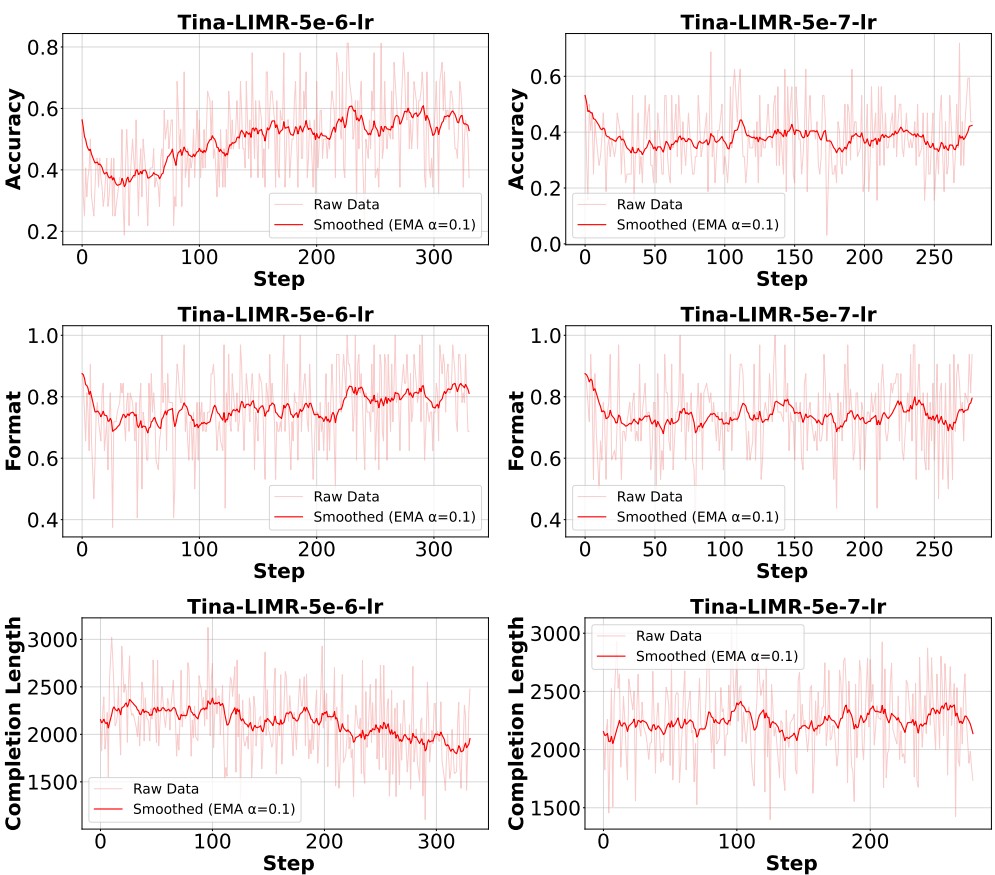

Figure 14: Transition in `Tina-LIMR-5e-6-lr` and `Tina-LIMR-5e-7-lr`.

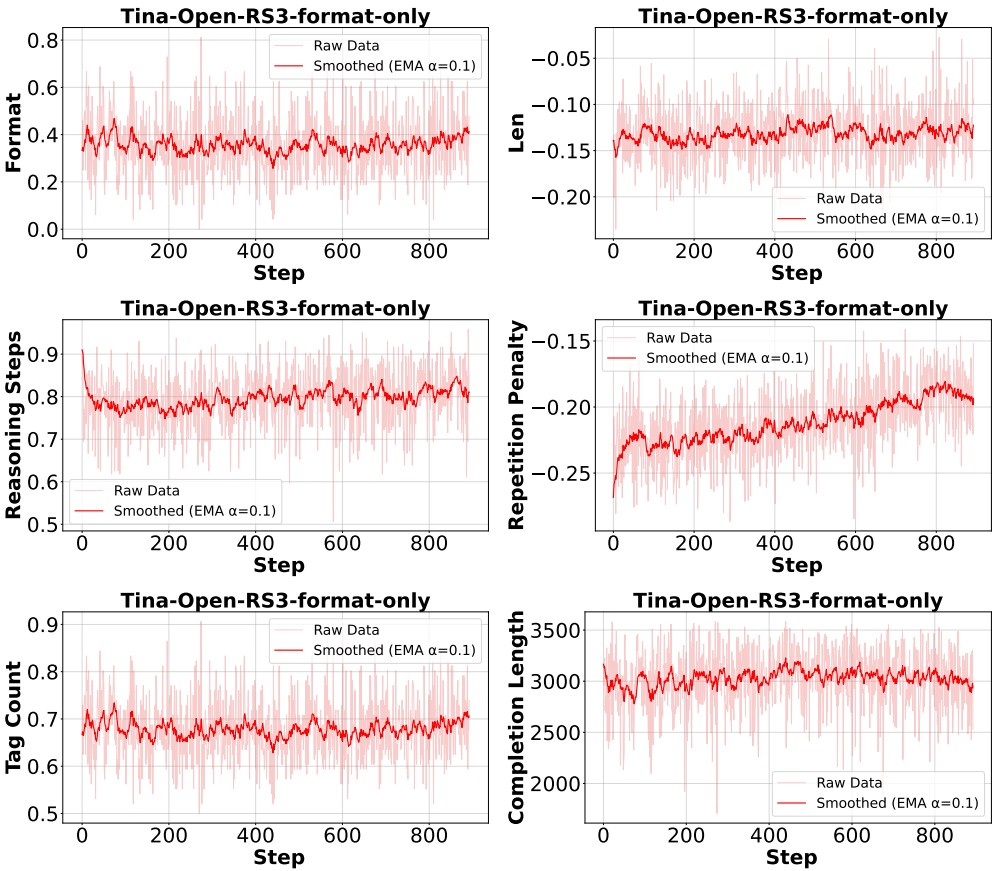

Figure 15: Transition in `Tina-Open-RS3-format-only`.

