# OpenReview forum: "Tina: Tiny Reasoning Models via LoRA"
_ICLR.cc/2026/Conference — ICLR 2026 Poster_

### Official Review · Reviewer_38rQ · 2025-10-26

**Soundness:** 3
**Presentation:** 4
**Contribution:** 3
**Rating:** 6
**Confidence:** 4

**Summary:**

This work proposes Tina -- a family of tiny reasoning models that achieve strong performance with extreme cost-efficiency offering a minimalist yet powerful path to efficient reasoning in small LMs. By applying LoRA during RL, Tina boosts reasoning ability at just $9 training cost which is very much cheaper than comparable models.

**Strengths:**

1. Novelty: One of the earliest works to demonstrate that GRPO based RL post-training can be used with LoRA
2. Cost-efficiency: Achieves SOTA or near SOTA reasoning performance at a tiny fraction of typical RL costs
3. Scalability & accessibility: Tiny models are easier to deploy and experiment with; open-source release ensures reproducibility
4. Comprehensive validation: Multiple datasets and ablations strengthen claims about effectiveness and generality

**Weaknesses:**

1. As mentioned in the paper, while the novelty is limited, theis is one of the few works to systematically investigate applying LoRA with GRPO for reasoning and while some tricks on training dynamic have been shared, the technical contribution looks limited.

**Questions:**

1. Authors, an ideas on extension of the work to tool-calling scenarios and multi-turn rollouts?

---

> ### Author Response · Authors · 2025-11-21
>
> We sincerely thank the reviewer for the thoughtful evaluation and for highlighting the key strengths of the work in cost-efficiency, accessibility, and comprehensive validation. We appreciate the recognition that demonstrating LoRA’s effectiveness within a GRPO-based RL setting is timely and valuable for researchers working with small models or limited compute. *We have also added a new Appendix C.3 in the modified paper, which extends Tina to larger model sizes and architectures (all updates in our revised paper are marked with blue text).*
>
> ### **On the scope of the technical contribution**
>
> We appreciate the reviewer’s comment regarding the simplicity of the method. We agree that our approach intentionally adopts a minimal combination of LoRA and GRPO, and we emphasize this explicitly in the paper. Our main aim is to investigate and understand an unexpectedly strong phenomenon.
>
> In this sense, the technical contribution lies in the scientific insights and analysis that emerge from this minimalist setup:
>
> - *The "Rapid Reasoning Format Adaptation" Hypothesis.* We propose and provide evidence for how LoRA succeeds in this domain. We hypothesize that it excels at rapidly adapting the model's output structure to the format rewarded by RL, while its parameter-efficient nature largely preserves the base model's underlying knowledge.
> - *The "Phase Transition" Dynamic.* Our analysis of the training dynamics (Section 6, Figure 3) identifies a consistent "phase transition" where format-related metrics (like length) destabilize, and the best-performing checkpoint consistently appears just before this point. This is a novel and actionable insight for training such models.
> - *Rigorous Dissection.* Our targeted ablations (Section 6.2) demonstrate that this phase transition is an emergent property of the interplay between accuracy and format rewards (disappearing in format-only training ) and not an artifact of token limits.
> We hope this analysis—which moves beyond just showing "what works" to providing a testable hypothesis and deep analysis for "why it works"—provides a useful contribution for the community, providing a practical and validated roadmap for "compute-scarce" research.
>
> ### **On extensions to tool-calling and multi-turn rollouts**
>
> We appreciate the reviewer’s forward-looking question. Both directions are interesting extensions of the work:
>
> - *Tool-calling.* Tool use requires the model to consistently generate structured, verifiable formats (e.g., API calls). Because the hypothesis suggests LoRA excels specifically at adapting output format under RL, we expect the method to extend naturally to tool-calling tasks, where format correctness can be verified and rewarded.
> - *Multi-turn rollouts.* Multi-turn reasoning and dialogue can be treated as longer-horizon rollouts where each turn contributes to a verifiable trajectory. GRPO can be adapted to reward multi-step behaviors, and LoRA’s knowledge-preserving nature is well-suited to maintaining coherence across turns. We see this as a promising next step and appreciate the reviewer for raising it.
>
> We thank the reviewer again for their positive assessment and constructive feedback. Their comments helped us clarify the contributions and highlight meaningful directions for future work.

---

### Official Review · Reviewer_N2DY · 2025-10-30

**Soundness:** 3
**Presentation:** 2
**Contribution:** 2
**Rating:** 2
**Confidence:** 3

**Summary:**

This paper proposes to use Low-rank adaptation(LORA) finetuning with a  group relative policy optimization (GRPO) approach in order achieve reasoning capacities with tiny models.

**Strengths:**

* Pursuits an interesting, potentially really important idea
* Easy to implement
* Large amount of experiments

**Weaknesses:**

The Idea is very straightforward (could also be seen as a plus!)

The paper is missing to report baseline benchmark results of small (but still larger models) for comparison (e.g., you could use something like a deepseek R1 model with ~7B parameters), e.g., in Table 2 and 3.; In the current state it is very hard to judge if the results could lead to any reasonable use case.

Measuring the training costs in $ is ephemeral and can change very quickly even without hardware changes (for economical reasons, exchange rates, etc...); For a scientific paper, I would recommend reporting FLOPs or at least GPU hours instead or on top.
Reporting the cost breakdown for different ablations and the evaluations to me seems not to be relevant for the main paper; Overall the main paper seems to be a bit stretched out to fill the pages and could be written way more concisely.

What am I missing is an in-depth discussion of the advantages and disadvantages of using very small models for reasoning and their place in an agentic AI. While I absolutely see reasons to use small models, reasoning appears to be *the* task, in which at least decently sized models seem to be a good investment;

The main paper is missing details about the actual training setup ("Tina models build upon the open-source training recipes and datasets
of STILL-3 (RUCAIBox STILL Team, 2025), DeepScaleR (Luo et al., 2025), and Open-RS (Dang and Ngo, 2025). " What does "build upon" mean here? Which datasets have been used? Is it a fair comparison if you just collect more datasets for fine-tuning?

**Questions:**

The paper goes into large efforts to show what can be achieved with small training costs.
An alternative to using tiny models could be to uses small models with quantization; How would these compare?

---

> ### Author Response · Authors · 2025-11-21
>
> We thank the reviewer for taking the time to provide detailed feedback and for highlighting the potential importance of the direction explored in this work. We appreciate the opportunity to clarify and strengthen the paper, and we are grateful for the reviewer’s suggestions, which helped us refine both the presentation and the scope of the experiments.
>
> The reviewer raised several helpful points regarding (1) larger-model baselines, (2) reporting FLOPs/GPU-hours in addition to price, (3) discussion of the role of small models in reasoning and agentic systems, and (4) clarity of the training setup. We have updated the paper to address each of these *(all updates in our revised paper are marked with blue text)*, and we also address each below.
>
> ### **Experiments with larger models**
>
> We agree with the reviewer that broader evidence across different model sizes and families is important. **Motivated by this suggestion, the revised version now includes a new Appendix C.3 that expands the experiments substantially beyond the original 1.5B base model.**
> In Appendix C.3.1, we evaluate LoRA-based GRPO against full-parameter GRPO across:
>
> - *Multiple model sizes.* 1.5B, 3B, 7B, and 14B sized models.
> - *Multiple model families.* Qwen2.5, Llama3.1-8B, and OctoThinker-3B
>
> The new results, summarized in Figures 6–10, show a consistent pattern: even with extremely small LoRA configurations (rank = 1), LoRA-based RL mirrors the reward dynamics of full-parameter GRPO across all tested models. We hope that this addresses the reviewer’s suggestion, and provides evidence that the observed phenomenon holds across multiple model sizes/families.
>
> ### **Reporting FLOPs and GPU hours**
>
> We appreciate the reviewer’s point about the importance of stable, hardware-agnostic metrics. Our intention in reporting dollar cost was to provide an intuitive sense of real-world accessibility, especially for researchers with limited budgets. That said, we agree that FLOPs and GPU-hours offer a more invariant view.
>
> To better reflect this, we have updated the paper with the blue text near the end of introduction and the beginning of Section 6.1, to emphasize the FLOPs/GPU-hour measurements, which are included in Figure 2 and Appendix B.1, and improve their prominence and clarity. We hope this gives readers both perspectives: a hardware-agnostic comparison and a concrete illustration of practical accessibility.
>
> ### **The role of small models**
>
> We appreciate the reviewer’s request for more discussion on the advantages and limitations of small models in the broader context of agentic systems. In response, we have added additional framing in the introduction section (please check the blue text in the modified introduction). In particular, we more clearly articulate that:
>
> - The goal of Tina is to explore the efficiency limits of RL-based reasoning and understand how far reasoning behavior can be pushed under tight compute budgets.
> - Many real-world agentic applications may benefit from lightweight, deployable models, and understanding their capabilities is scientifically valuable.
> - Importantly, as now shown in Appendix C.3.1, the Tina methodology is not restricted to tiny models—the same trends hold when applied to larger 3B–14B models. We hope this provides a clearer picture of where small models fit relative to larger models and how the method extends across scales.
>
> ### **Training setup and data sources**
>
> We thank the reviewer for pointing out that the description could be clearer. To address this, we have revised Section 4.1, Appendix B.2 and B.3 to more explicitly give details about our setup. In particular, Section 4.1 describes the overall baselines, training datasets, and evaluation benchmarks. Appendix B.2 describes in detail the training datasets composition, reward setup, and hyperparameter inherited from training recipes of baseline models including STILL-3, DeepScaleR, and Open-RS. Appendix B.3 gives details about the benchmark tasks, GPU infra, codebase. Our goal was to ensure that the experimental pipeline is easy to understand and fully reproducible, and we appreciate the reviewer’s suggestion to clarify this further.
>
> ### **On quantization**
>
> This is an excellent direction, and we thank the reviewer for raising it. **In our updated paper, we have also added new experiments in Appendix C.3.2 evaluating QLoRA and QDoRA variants, which combine our RL setup with 4-bit base-model quantization.** These results show that quantized LoRA variants behave similarly to the full-precision LoRA case, suggesting that the approach is compatible with and complementary to quantization techniques.
>
> We sincerely thank the reviewer again for the constructive comments. These suggestions substantially improved the clarity, scope, and presentation of the paper.

---

### Official Review · Reviewer_XvES · 2025-11-07

**Soundness:** 4
**Presentation:** 3
**Contribution:** 3
**Rating:** 8
**Confidence:** 4

**Summary:**

This paper presents a study on using LoRA and RL to improve the reasoning capability of small language models. Through extensive experiments, the authors show us a surprising conclusion that LoRA, compared to full-parameter tuning, is more efficient and effective when using RL to improve the reasoning capability of small language models. To explain this phenomenon, the authors propose the Rapid Reasoning Format Adaptation Hypothesis, suggesting that LoRA enables models to quickly adapt to reasoning formats rewarded by RL while largely preserving the base model’s knowledge. They conduct rigorous ablation studies and detailed training-dynamics analyses to support this claim. Overall, the paper is well-organized, technically sound, and reproducible, offering clear insights into the mechanisms of parameter-efficient RL and setting a valuable precedent for future research.

**Strengths:**

1. This paper is well written and clearly presents the necessary background, experimental setup, and extensive ablation studies that support the authors’ claims.

2. The experimental setup is clean and effective. The paper adopts a minimalist yet powerful combination of LoRA and GRPO reinforcement learning, eliminating noise from architectural or dataset improvements, which is essential and helps focus on the most important research question the paper aims to answer.

3. The authors propose the Rapid Reasoning Format Adaptation Hypothesis to explain why they observe that RL with LoRA is more effective for reasoning, and rigorous ablation studies support this. This is very beneficial for the entire research community.

**Weaknesses:**

1. Although the paper evaluates Tina across multiple training recipes (STILL-3-1.5B, DeepScaleR-1.5B, and Open-RS), all experiments ultimately rely on the same underlying base model, DeepSeek-R1-Distill-Qwen-1.5B. This raises concerns about the generality of the claimed findings. It remains unclear whether the observed “rapid reasoning adaptation” and cost-efficiency would still hold for other base models that are not distilled from DeepSeek-R1 or outside the Qwen2.5 family. For instance, would similar trends be observed for models such as Qwen3-1.7B or other architectures of comparable scale? A broader evaluation would strengthen the paper’s generality.

2. The authors attribute LoRA’s effectiveness to its ability to rapidly adapt the reasoning format under RL while preserving the base model’s knowledge. However, using a sufficiently small learning rate in full-parameter fine-tuning could potentially yield similar effects: slow, stable updates that preserve existing knowledge while adapting reasoning behavior. It would be valuable to analyze whether such small-learning-rate full-parameter fine-tuning exhibits the same adaptation dynamics or efficiency gains as LoRA, and to clarify the conceptual and empirical differences between the two.

**Questions:**

See Weaknesses.

---

> ### Author Response · Authors · 2025-11-21
>
> We thank the reviewer for their thoughtful assessment. We appreciate the recognition of the paper’s clarity, technical soundness, and reproducibility, as well as the reviewer’s comments on the value of the minimalist experimental design and the proposed Rapid Reasoning Format Adaptation Hypothesis. We are grateful for the insightful suggestions, which helped us strengthen the paper.
>
> The reviewer raises two helpful questions regarding (1) generality beyond the single base model, and (2) the comparison between full-parameter tuning and LoRA. We thank the reviewer for identifying these important points, and we have updated the paper to address them more thoroughly *(all updates in our revised paper are marked with blue text).*
>
> ### **On the generality beyond the single base model**
>
> We agree with the reviewer that broader evidence across different families and scales is important. Motivated by this suggestion, the revised version now includes a new Appendix C.3 that expands the experiments substantially beyond the original 1.5B base model. In Appendix C.3.1, we evaluate LoRA-based GRPO against full-parameter GRPO across:
>
> - *Multiple model sizes.* 1.5B, 3B, 7B, and 14B sized models.
> - *Multiple model families.* Qwen2.5, Llama3.1-8B, and OctoThinker-3B
>
> The new results, summarized in Figures 6–10, show a consistent pattern: even with extremely small LoRA configurations (rank = 1), LoRA-based RL mirrors the reward dynamics of full-parameter GRPO across all tested models. We hope that this addresses the reviewer’s suggestion, and provides evidence that the observed phenomenon is not specific to one specific base model.
>
> ### **On full-parameter tuning (low learning rate) vs. LoRA**
>
> We also appreciate the reviewer’s question regarding whether small-learning-rate full-parameter fine-tuning might exhibit similar behavior. This is an insightful direction, and we have clarified this distinction in the revision.
>
> Our findings suggest two key differences:
>
> - *Practical feasibility.* Full-parameter RL—even with small learning rates—requires storing and updating optimizer states and gradients for all parameters. Under the minimal hardware setup used in this paper, this is often not feasible in practice. LoRA’s parameter-efficient structure is what makes the “tiny-cost” setting possible.
> - *Empirical behavior.* The reviewer’s intuition about small learning rates yielding stable adaptation is reasonable, but our experiments indicate that effective RL adaptation in this setting consistently requires higher learning rates for the trainable components. In Appendix C.3.2, LoRA-based variants use learning rates that are 20× larger than the full-parameter GRPO baseline. In our LoRA learning-rate ablations (Table 3), the smallest learning rates underperform. That being said, we hope to more thoroughly explore behavior of full-parameter updates with respect to learning rates in future work.
>
> These results suggest that LoRA’s advantage does not stem from simulating slow full-parameter updates. Instead, the structural constraint of freezing the base model enables fast, targeted adaptation with learning rates that would destabilize full-parameter training.
>
> We again thank the reviewer for their careful evaluation and constructive feedback. Their comments motivated several additions that we believe significantly strengthen the generality and clarity of the paper.

---

### Official Review · Reviewer_au9R · 2025-11-08

**Soundness:** 3
**Presentation:** 4
**Contribution:** 2
**Rating:** 6
**Confidence:** 3

**Summary:**

This paper introduces TINA, demonstrating that applying LoRA during reinforcement learning to a tiny 1.5B model can achieve SOTA-competitive reasoning performance at a tiny fraction of the computational cost.

**Strengths:**

- The authors' research finding that LoRA on small language models can achieve performance on par with full-parameter RL training is highly valuable. I believe this will benefit researchers who are "compute-scarce" and those who are new to the field, allowing more people to experiment with and master RL scaling.

- The hypothesis proposed by the authors makes sense. It holds similar viewpoints with the concurrent blog-"LoRA Without Regret", suggesting that the amount of information gained during the RL process is less than that in SFT. Therefore, a parameter-efficient method like LoRA can be effectively used to "organize" the base model's existing knowledge into the "structured format" preferred by a RL reward scalar.

**Weaknesses:**

Please correct me if my understanding is wrong or biased:-)

Overall, TIna is a very "timely" paper. Its primary contribution does not lie in proposing a new methodology, contributing new data or benchmarks, or releasing a powerful SOTA model. Instead, its main contribution comes from "updating the reader's cognition." The authors are loudly proclaiming to the community that "LoRA + small language models can still achieve significant reasoning gains via RL," essentially encouraging everyone to "Go try LoRA RL training now!" If I had reviewed this paper earlier this year for a conference like COLM or NeurIPS, I would have given it a clear accept. However, at this current point in time, I feel compelled to raise two additional questions:

- Could the authors provide deeper insights? For example, whether other PEFT methods (e.g., DoRA), or different model families (e.g., Llama, MiMO, or even VLMs), or non-mathematical reasoning tasks (e.g., puzzles, games) are applicable to TINA? Or perhaps some deeper theoretical insights?

- LoRA RL Training is already supported by several existing RL frameworks, such as verl, and they are often scalable, supporting training on larger models and more tasks. Given this, what do the authors believe is the unique, standalone value of the open-sourced code and model weights provided by TINA?

**Questions:**

The first citation in this paper appears to be sensitive. It indirectly points to a blog post that is directly related to this work. I am not sure if this potentially violates the double-blind review process.

**Details Of Ethics Concerns:**

nan

---

> ### Author Response · Authors · 2025-11-21
>
> We sincerely appreciate the reviewer’s thoughtful and encouraging assessment. We are grateful for the recognition of this work’s value to compute-constrained researchers and for the reviewer’s positive remarks on the clarity of the hypothesis.
>
> The reviewer raises two helpful questions regarding (1) deeper insights and generality, and (2) the standalone value of our released artifacts. We thank the reviewer for identifying these important points, and we have updated the paper to address them more thoroughly *(all updates in our revised paper are marked with blue text).*
>
> ### **On deeper insights and generalizability**
>
> We fully agree that understanding the broader applicability of the phenomenon is essential. Motivated by this suggestion, the revised version now includes a new Appendix C.3 with additional experiments that extend the scope of our findings:
>
> - *Across model families and scales.* In Appendix C.3.1, we apply LoRA-based GRPO to Qwen2.5-3B/7B/14B, Llama3.1-8B, and OctoThinker-3B (mid-trained on Llama3.2-3B). As shown in Figures 6–10, the same behavior holds across architectures and sizes, even when LoRA rank is as small as 1. This reflects the reviewer’s suggestion to explore broader families and confirms that the effect is not limited to the original 1.5B setting.
> - *Across PEFT variants.* In Appendix C.3.2, we evaluate DoRA, QLoRA, and QDoRA. Figure 11 shows that each variant behaves similarly to full-parameter GRPO. This directly responds to the reviewer’s question about alternative PEFT methods.
>
> We hope these new results meaningfully strengthen the paper's insight into generality.
>
> ### **On the unique value of the Tina open-source release**
>
> We also understand the reviewer’s point regarding existing RL frameworks that support LoRA. Our intent is not to introduce another framework, but to provide:
>
> - a fully reproducible reference implementation tailored to the specific phenomenon studied in this paper,
> - a set of minimal-compute training logs, weights, and checkpoints corresponding exactly to the results reported, and
> - a transparent and lightweight pipeline that readers can easily run on two GPUs to reproduce every experiment, including all ablations.
>
> We agree that, in an ecosystem with many RL frameworks, clear and complete reproducibility artifacts remain important. We thank the reviewer for prompting us to clarify this contribution more explicitly.
>
> ### **On the first citation**
>
> Thanks for your careful attention here. Out of an abundance of caution and to avoid any ambiguity, we have updated the citation in question to refer to a neutral, archival source.
>
> We thank the reviewer again for their constructive comments, which helped us improve the clarity, scope, and presentation of the paper.

---

### Author Response · Authors · 2025-12-03
**Summary of Revisions and Responses**

We thank the reviewers for their constructive feedback and the generally positive reception (Scores: 8, 6, 6, 2). The reviewers highlighted the work’s timeliness, reproducibility, and high value to the "compute-scarce" research community (Reviewer au9R), calling our findings on LoRA efficiency "potentially really important" (Reviewer XvES) and the experimental validation "comprehensive" (Reviewer 38rQ).

We summarize here how our rebuttal and revised manuscript have addressed the key concerns raised, particularly regarding the generality of our findings and the concerns of the dissenting reviewer.

### **Generality and Scaling (Addressing the Main Critique)**

The primary concern shared by the lower-scoring Reviewer N2DY (also asked by Reviewers XvES and au9R), was whether our findings hold beyond the 1.5B parameter scale and the specific Qwen-distilled base model.

- *New Experiments (Appendix C.3.1)*: We have added extensive experiments applying Tina’s LoRA-based GRPO to multiple model sizes (3B, 7B, and 14B sized models), as well as multiple model families (Qwen2.5, Llama3.1 and OctoThinker).
- *Results*: The findings remain consistent across all model families and scales: LoRA-based RL achieves reasoning performance comparable to full-parameter training while maintaining superior training stability and efficiency.

### **Addressing Reviewer N2DY (Score: 2)**

Reviewer N2DY acknowledged the idea is "potentially really important" but raised specific concerns regarding baselines, metrics, and training details. We believe each of these have been fully addressed:

- *Concern: "Missing larger model baselines (e.g., ~7B)."* Response: As noted above, we have added comparisons for 3B, 7B, 8B, and 14B models. The results confirm that our conclusions are not limited to tiny models.
- *Concern: "Training costs in $ is ephemeral... recommend FLOPs."* Response: We agree. While we used dollar amounts for accessibility, FLOPs were already included in Figure 2 and Appendix B.1. We have revised the text to prominently feature FLOPs and GPU-hours as the primary invariant metrics.
- *Concern: "Missing details about actual training setup."* Response: We have revised Section 4.1 and Appendix B to explicitly detail these configurations. Furthermore, we provide full code and training logs for exact reproducibility. We also note that we utilize standard open-source recipes (STILL-3, Open-RS).
- *Concern: "Comparison with quantization."* Response: We have also added Appendix C.3.2, evaluating QLoRA and QDoRA (4-bit quantization). The results show that our method works seamlessly with quantization, further enhancing efficiency.

---

### Meta-Review · Area_Chair_EBDV · 2026-01-07

**Summary:**

This paper introduces TINA, a family of cost-efficient reasoning models showing that applying LoRA during reinforcement learning can achieve reasoning performance comparable to full-parameter RL while using a tiny fraction of the compute. Through extensive experiments and analyses, the authors demonstrate that parameter-efficient RL rapidly adapts the model’s reasoning format without overwriting existing knowledge, a phenomenon they formalize as the Rapid Reasoning Format Adaptation Hypothesis. The results hold across multiple model sizes, architectures, and PEFT variants, highlighting a practical and accessible path to reasoning improvements for compute-constrained settings.

The reviewers broadly agree that the paper presents a timely and carefully executed study showing that applying LoRA during reinforcement learning can yield reasoning performance competitive with full-parameter RL, at dramatically lower computational cost. This finding is considered valuable for compute-constrained researchers and for improving understanding of how RL interacts with parameter-efficient fine-tuning. Strengths highlighted include clean experimental design, extensive ablations, strong cost-efficiency, and the proposal of the Rapid Reasoning Format Adaptation hypothesis supported by training-dynamics analyses.

The main reservations concerned (i) the novelty and depth of contribution relative to the maturity of the field, (ii) the initial lack of evidence for generality across model families, PEFT methods, and scales, (iii) clarity around baselines, training setup, and cost reporting, and (iv) concerns about double-blind compliance due to an early citation **(need SAC and PC to double check this violating the double-blind policy)**

The paper provides a well-substantiated, reproducible, and practically meaningful insight into parameter-efficient reinforcement learning, with rebuttals that substantially address concerns about generality, clarity, and experimental scope. The empirical evidence and analysis collectively justify acceptance as a solid contribution with clear value to the community.

**NOTE: (need SAC and PC to double-check this, violating the double-blind policy)**

"Reviewer N2DY
I think that the double-blindness of this paper is borderline. I just followed the first link in the references (which is also the first citation in the paper), and more or less directly landed on a non-blind blog post about this paper."

**Reviewer Concerns:**

**Concerns largely addressed by the rebuttal and revision:**

Generality across models and methods: Additional experiments across multiple model families (e.g., Qwen, Llama, OctoThinker), scales (1.5B–14B), and PEFT variants (DoRA, QLoRA, QDoRA) substantially strengthen the claim that the observed phenomenon is not model- or method-specific.

Comparison to alternative explanations: The discussion contrasting LoRA with low–learning-rate full-parameter RL clarifies both practical feasibility and empirical differences.

Baseline clarity and training details: Expanded appendices and clarifications on datasets, training recipes, and evaluation improve reproducibility and transparency.

Cost reporting: Greater emphasis on FLOPs and GPU-hours alongside dollar cost addresses concerns about the stability of cost metrics.

Concerns that remain partially outstanding:

Some reviewers may still view the core idea as conceptually simple and incremental, despite the strengthened empirical evidence and analysis.

Impact beyond the studied setting: Extensions to more complex scenarios (e.g., tool use, multi-turn rollouts) remain speculative rather than demonstrated.

**Reviewer Scores:**

Reviewer au9R: Likely to maintain the score (≈6), given that requests for broader applicability, PEFT variants, and citation issues were directly addressed.

Reviewer XvES: Likely to maintain a strong accept-level score (≈8 → 8), as the main concerns about generality and LoRA vs. full-parameter tuning were comprehensively handled.

Reviewer N2DY: May increase slightly but likely remain negative or borderline (≈2 –4), as concerns about novelty, framing, and use-case relevance are only partially alleviated.

Reviewer 38rQ: Likely to maintain a marginally positive score (≈6), with clarifications strengthening the contribution but not fundamentally changing views on limited novelty.

---

### Decision · Program_Chairs · 2026-01-26

Accept (Poster)